# Statistical Test for Diffusion-Based Anomaly Localization via Selective Inference

## Abstract

Anomaly localization in images—identifying regions that deviate from normal patterns—is vital in applications such as medical diagnosis and industrial inspection. A recent trend is the use of image generation models in anomaly localization, where these models generate normal-looking counterparts of anomalous images, thereby allowing flexible and adaptive anomaly localization. However, these methods inherit the uncertainty and bias implicitly embedded in the employed generative model, raising concerns about the reliability. To address this, we propose a statistical framework based on selective inference to quantify the significance of detected anomalous regions. Our method provides $p$-values to assess the false positive detection rates, providing a principled measure of reliability. As a proof of concept, we consider anomaly localization using a diffusion model and its applications to medical diagnoses and industrial inspections. The results indicate that the proposed method effectively controls the risk of false positive detection, supporting its use in high-stakes decision-making tasks.

## 1 Introduction

Anomaly localization using image generation models, particularly diffusion models, has shown great promise across diverse domains such as medical diagnosis and industrial inspection (Li et al., 2023; Iqbal et al., 2023; Lu et al., 2023; Zhang et al., 2023; Fontanella et al., 2024; Tebbe & Tayyub, 2024; Sheng et al., 2024). These models reconstruct *a normal-looking version* of an input image, and differences between the input and the reconstruction highlight potential anomalies. Compared to traditional anomaly localization methods, generative approaches are highly suitable for settings where annotations for anomalous regions are unavailable. Moreover, generative approaches can flexibly handle heterogeneity by adapting to individual images—e.g., patient-specific characteristics in medical diagnosis and product-specific traits in industrial inspection. Among various image generation models, diffusion models, in particular, offer high fidelity and stability, often outperforming other methods in image quality and anomaly localization.

While generative approaches offer powerful and flexible capabilities for anomaly localization, a major concern is that the inherent uncertainty and bias in generative models can affect localization performance (Fithian et al., 2014; Taylor & Tibshirani, 2015; Lee et al., 2016; Duy & Takeuchi, 2022; Miwa et al., 2023; Shiraishi et al., 2024b). These models are trained on specific datasets composed of normal images, and the quality of the generated normal-looking images depends heavily on the dataset distribution and how accurately the model has learned it. As a result, uncertainties or biases in the dataset or training process can cause incorrect reconstructions, leading to inaccurate localizations and misidentification of anomalies. Such risks are especially critical in high-stakes domains such as medical diagnosis and industrial inspection, where even minor errors can have serious consequences. Therefore, it is essential to incorporate a rigorous uncertainty quantification framework and statistical safeguards to ensure reliable deployment in critical applications.

To address this issue, we propose a statistical testing framework based on Selective Inference (SI) to assess the statistical significance of detected anomalies. SI has emerged as a promising approach for conducting statistical inference on hypotheses that are selected based on observed data. In this framework, inference is performed using the conditional distribution given the selection event, thereby accounting for the uncertainty

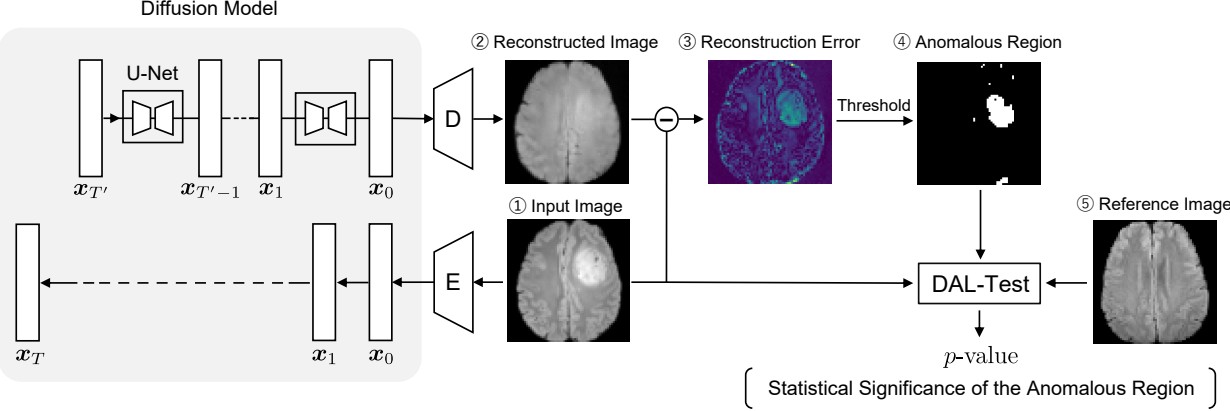

Figure 1: Schematic illustration of anomaly localization using a diffusion model and the proposed DAL-Test. When a test image—potentially containing an anomalous region—is fed into a trained diffusion model, its normal-looking version is generated through the forward and reverse processes. By comparing the input image with the generated normal-looking version, the anomalous region can be identified. We propose the *Diffusion-based Anomaly Localization (DAL) Test*, which leverages the selective inference framework to compute valid $p$-values that quantify the statistical significance of anomalous regions detected by a diffusion model, based on a test statistic defined over the input and reference images.

and bias associated with the hypothesis selection. Following this SI framework, our key idea is to apply statistical testing to detected anomalies, conditioned on the fact that the anomalous regions were identified using a specific generative model. This allows us to quantify the statistical significance of detected anomalies as a valid $p$-value, providing a rigorous estimate of the false positive rate and offering a principled metric for reliability. The overall framework of the proposed method is illustrated in Figure 1.

In this paper, as a proof of concept for the proposed statistical testing framework, we focus on a standard *denoising diffusion probabilistic model (DDPM)* (Ho et al., 2020; Song et al., 2022) among various diffusion-based anomaly detection (AD) methods, and demonstrate its applications to medical diagnostics and industrial inspections. However, the proposed framework is readily generalizable to a broader range of diffusion model architectures and is applicable to semi-supervised AD problems in other domains.

**Related works**  Diffusion models have been effectively utilized in anomaly localization problems (Pinaya et al., 2022; Fontanella et al., 2024; Wyatt et al., 2022; Mousakhan et al., 2023). In this context, the *DDPM* is commonly used (Ho et al., 2020; Song et al., 2022). During the training phase, a DDPM model learns the distribution of normal medical images by iteratively adding and then removing noise. In the test phase, the model attempts to reconstruct a new test image. If the image contains anomalous regions, such as tumors, the model may struggle to accurately reconstruct these regions, as it has been trained primarily on normal regions. The discrepancies between the original and the reconstructed image are then analyzed to identify and highlight anomalous regions. Other types of generative models have also been used for the anomaly localization task (Baur et al., 2021; Chen & Konukoglu, 2018; Chow et al., 2020; Jana et al., 2022).

SI was first introduced within the context of reliability evaluation for linear model features when they were selected using a feature selection algorithm (Lee & Taylor, 2014; Lee et al., 2016; Tibshirani et al., 2016), and then extended to more complex feature selection methods (Yang et al., 2016; Suzumura et al., 2017; Hyun et al., 2018; Rügamer & Greven, 2020; Das et al., 2021). Then, SI proves valuable not only for feature selection problems but also for statistical inference across various data-driven hypotheses, including unsupervised learning tasks (Chen & Bien, 2020; Tsukurimichi et al., 2021; Tanizaki et al., 2020; Duy et al., 2022; Le Duy et al., 2024; Lee et al., 2015; Gao et al., 2022; Duy et al., 2020; Jewell et al., 2022). The fundamental idea of SI is to perform inference conditional on the hypothesis selection event, which mitigates the selection bias issue even when the hypothesis is selected and tested using the same data. To conduct SI, it is necessary to derive the sampling distribution of the test statistic conditional on the hypothesis selection event. To the

best of our knowledge, SI was applied to statistical inferences on several deep learning models (Duy et al., 2022; Miwa et al., 2023; Shiraishi et al., 2024b; Miwa et al., 2024), but none of these works addresses image generation by diffusion models. In particular, the closest work to ours assigns SI $p$-values to anomalous regions detected by VAEs (Miwa et al., 2024), where the detector is a single deterministic encoder–decoder pass and the selection event is thus characterized through a single application of the network. In contrast, a diffusion model detects the region through a recursive and stochastic generation process composed of noising operations and many U-Net evaluations over multiple time steps, so the selection event is induced by this entire trajectory, including the injected noises. Characterizing this event does not follow from the single-pass case and constitutes the main technical contribution of this work.

**Contributions**  The main contributions of our study are summarized as follows[1].

- We propose a novel statistical testing framework to assess the significance of anomaly localization results obtained from diffusion model-based methods, offering a principled basis for evaluating the reliability of detected anomalies.

- We implement the SI framework for diffusion models by deriving the sampling distribution conditional on the selection event induced by the diffusion model, which requires developing non-trivial computational techniques tailored to the generative sampling process.

- We provide theoretical justification for the proposed method and validate its effectiveness through extensive numerical experiments in medical diagnosis and industrial inspection scenarios. The results highlight the robustness and practical utility of our method.

The implementation code for reproducing all experimental results is provided as supplementary material.

## 2  Anomaly localization by diffusion models

This section describes the anomaly localization task based on a diffusion model, which is explored as a proof of concept in this study. The process of anomaly localization using generative models can generally be divided into two phases. First, during the training phase, a denoising diffusion probabilistic model (DDPM) is trained using a dataset composed exclusively of normal images. The model learns the distribution of normal images through two key processes: the diffusion process, in which noise is gradually added to an image, and the reverse diffusion process, in which the original image is reconstructed from noise. Through this procedure, the model enhances its capacity to reconstruct normal image structures by acquiring denoising capabilities at each step. Next, during the testing phase, the reverse diffusion process is conditionally applied to an unseen input test image. In this step, the model reconstructs an image that closely resembles the input but conforms to its learned notion of "normality", causing anomalous regions to be poorly reproduced. An anomaly score is then computed based on the difference between the reconstructed image and the input test image, and the spatial distribution of this score is analyzed to localize anomalies. By applying thresholding to the score map, anomalous regions can be identified.

In this study, for the purpose of proof of concept, we adopt standard denoising diffusion models as our choice of diffusion model (Ho et al., 2020; Song et al., 2022). We outline the image reconstruction process of a trained DDPM below. Given a test image that possibly contains anomalous regions, a denoising diffusion model is used to generate the corresponding normal image. The reconstruction consists of two processes called *forward process (or diffusion process)* and *reverse process*. In the forward process, noise is sequentially added to the test image so that it converges to a standard Gaussian distribution $\mathcal{N}(\mathbf{0}, I)$. Let $\boldsymbol{x}$ be an image represented as a vector with each element corresponding to a pixel value. Given an original test image $\boldsymbol{x}_0$, noisy images $\boldsymbol{x}_1, \boldsymbol{x}_2, \ldots, \boldsymbol{x}_T$ are sequentially generated, where $T$ is the number of noise addition steps. We consider the distribution of the original and noisy test images, which is denoted by $q(\boldsymbol{x})$, and approximate the

---

[1]We note that our contribution is *not* the development of a new diffusion-based anomaly localization algorithm, but rather the introduction of a rigorous statistical testing framework designed to quantify the statistical reliability of anomalous regions identified by diffusion-based AD methods.

distribution by a parametric model $p_\theta(\boldsymbol{x})$ with $\theta$ being the parameters. Using a sequence of noise scheduling parameters $0 < \beta_1 < \beta_2 < \cdots < \beta_T < 1$, the forward process is written as

$$q(\boldsymbol{x}_{1:T}|\boldsymbol{x}_0) := \prod_{t=1}^{T} q(\boldsymbol{x}_t|\boldsymbol{x}_{t-1}), \quad \text{where} \quad q(\boldsymbol{x}_t|\boldsymbol{x}_{t-1}) := \mathcal{N}(\sqrt{1-\beta_t}\boldsymbol{x}_{t-1}, \beta_t I).$$

By the reproducibility of the Gaussian distribution, $\boldsymbol{x}_t$ can be rewritten by a linear combination of $\boldsymbol{x}_0$ and $\epsilon_t$, i.e.,

$$\boldsymbol{x}_t = \sqrt{\alpha_t}\boldsymbol{x}_0 + \sqrt{1-\alpha_t}\epsilon_t, \quad \text{with} \quad \epsilon_t \sim \mathcal{N}(\boldsymbol{0}, I), \tag{1}$$

where $\alpha_t = \prod_{s=1}^{t}(1-\beta_s)$. In the reverse process, a parametric model in the form of $p_\theta(\boldsymbol{x}_{t-1}|\boldsymbol{x}_t) = \mathcal{N}(\mu_\theta(\boldsymbol{x}_t, t), \beta_t I)$ is employed, where $\mu_\theta(\boldsymbol{x}_t, t)$ is obtained by using the predicted noise component $\epsilon_\theta^{(t)}(\boldsymbol{x}_t)$. Typically, a U-Net is used as the model architecture for $\epsilon_\theta^{(t)}(\boldsymbol{x}_t)$. In DDPM (Ho et al., 2020), the loss function for training the noise component is simply written as $||\epsilon_\theta^{(t)}(\boldsymbol{x}_t) - \epsilon_t||_2^2$. Based on (1), given a noisy image $\boldsymbol{x}_t$ after $t$ steps, the reconstruction of the image in the previous step $\boldsymbol{x}_{t-1}$ is obtained as

$$\boldsymbol{x}_{t-1} = \sqrt{\alpha_{t-1}} \cdot f_\theta^{(t)}(\boldsymbol{x}_t) + \sqrt{1-\alpha_{t-1}-\sigma_t^2} \cdot \epsilon_\theta^{(t)}(\boldsymbol{x}_t) + \sigma_t \epsilon_t, \tag{2}$$

where

$$f_\theta^{(t)}(\boldsymbol{x}_t) := (\boldsymbol{x}_t - \sqrt{1-\alpha_t} \cdot \epsilon_\theta^{(t)}(\boldsymbol{x}_t))/\sqrt{\alpha_t}, \tag{3}$$

and

$$\sigma_t = \eta\sqrt{(1-\alpha_{t-1})/(1-\alpha_t)}\sqrt{1-\alpha_t/\alpha_{t-1}}. \tag{4}$$

Here, $\eta$ is a hyperparameter that controls the randomness in the reverse process. By setting $\eta = 1$, we can create new images by stochastic sampling. On the other hand, if we set $\eta = 0$, deterministic sampling is used for image generation. By recursively sampling as in (2), we can obtain a reconstructed image of the original input $\boldsymbol{x}_0$.

In practice, the reverse process starts from $\boldsymbol{x}_{T'}$ with $T' < T$. Namely, we reconstruct the normal-looking version of the input image not from a completely noisy one, but from one that still contains individual information of the original input image. The smaller $T'$ ensures that the reconstructed image preserves fine details of the input image. Conversely, the larger $T'$ results in the retention of only large-scale features, thereby converting more of the anomalous regions into normal regions (Ho et al., 2020; Mousakhan et al., 2023). The overall reconstruction process of the diffusion model is summarized in Algorithm 1.

---

**Algorithm 1** Reconstruction Process

**Require:** Input image $\boldsymbol{x}$
1: $\boldsymbol{x}_{T'} \leftarrow \sqrt{\alpha_{T'}}\boldsymbol{x} + \sqrt{1-\alpha_{T'}}\epsilon$
2: **for** $t = T', \ldots, 1$ **do**
3: $\quad f_\theta^{(t)}(\boldsymbol{x}_t) \leftarrow (\boldsymbol{x}_t - \sqrt{1-\alpha_t} \cdot \epsilon_\theta^{(t)}(\boldsymbol{x}_t))/\sqrt{\alpha_t}$
4: $\quad \boldsymbol{x}_{t-1} \leftarrow \sqrt{\alpha_{t-1}} \cdot f_\theta^{(t)}(\boldsymbol{x}_t) + \sqrt{1-\alpha_{t-1}-\sigma_t^2} \cdot \epsilon_\theta^{(t)}(\boldsymbol{x}_t) + \sigma_t \epsilon_t$
5: **end for**
**Ensure:** Reconstructed image $\boldsymbol{x}_0$

---

## 3 Statistical test for diffusion-based anomaly localization

In this section, we formulate the statistical test for detecting anomalous regions by a trained diffusion model. As shown in Figure 1, anomalous region detection by diffusion models is conducted as follows. First, in the training phase, the diffusion model is trained only on normal images. Then, in the test phase, we feed a test image that might contain anomalous regions into the trained diffusion model, and reconstruct it back from a noisy image $\mathbf{x}_{T'}$ at step $T' < T$. By appropriately selecting $T'$, we can generate a normal-looking image that retains individual characteristics of the test input image. If the image does not contain anomalous regions, the reconstructed image is expected to be similar to the original test image. On the other hand, if the image

contains anomalous regions, such as tumors, the model may struggle to reconstruct these regions, as it has been trained primarily on normal regions. Therefore, the anomalous regions can be detected by comparing the original test image and its reconstructed one.

**Problem formulation**   We develop a statistical test to quantify the reliability of decision-making based on images generated by diffusion models. To develop a statistical test, we interpret an image as a sum of a true signal component $\boldsymbol{\mu} \in \mathbb{R}^n$ and a noise component $\boldsymbol{\varepsilon} \in \mathbb{R}^n$. We emphasize that the noise component $\boldsymbol{\varepsilon}$ should not be confused with the noise $\epsilon$ used in the forward process. No structural assumption is imposed on the signal $\boldsymbol{\mu}$. On the other hand, regarding the noise component, it is assumed to follow a Gaussian distribution, and its covariance matrix is estimated using normal data different from that used for the training of the diffusion model, which is the standard setting of SI. Namely, an image with $n$ pixels can be represented as an $n$-dimensional random vector,

$$\boldsymbol{X} = (X_1, X_2, \ldots, X_n)^\top = \boldsymbol{\mu} + \boldsymbol{\varepsilon}, \quad \boldsymbol{\varepsilon} \sim \mathcal{N}(\boldsymbol{0}, \Sigma),$$

where $\boldsymbol{\mu} \in \mathbb{R}^n$ is the unknown true signal vector and $\Sigma$ is the covariance matrix. In the following, we use $\boldsymbol{X}$ to emphasize that an image is considered as a random vector, while the observed image vector is denoted as $\boldsymbol{x}$.

Let us denote the reconstruction process of the trained diffusion model in Algorithm 1 as the mapping from an input image to the reconstructed image $\mathcal{D} \colon \mathbb{R}^n \ni \boldsymbol{X} \to \mathcal{D}(\boldsymbol{X}) \in \mathbb{R}^n$. The difference between the input image $\boldsymbol{X}$ and the reconstructed image $\mathcal{D}(\boldsymbol{X})$ indicates the reconstruction error. When identifying anomalous regions based on reconstruction error, it is useful to apply some filter to remove the influence of pixel-wise noise. In this study, we use an averaging filter. Let us denote the averaging filter as $\mathcal{F} \colon \mathbb{R}^n \to \mathbb{R}^n$. Then, the process of obtaining the (filtered) reconstruction error is written as

$$\mathcal{E} \colon \mathbb{R}^n \ni \boldsymbol{X} \mapsto |\mathcal{F}(\boldsymbol{X} - \mathcal{D}(\boldsymbol{X}))| \in \mathbb{R}^n,$$

where the absolute value is taken pixel-wise. Anomalous regions are then obtained by applying a threshold to the filtered reconstruction error $\mathcal{E}_i(\boldsymbol{X})$ for each pixel $i \in [n]$. Specifically, we define the anomalous region as the set of pixels whose filtered reconstruction error is greater than a given threshold $\lambda \in \mathbb{R}^+$, i.e.,

$$\mathcal{M}_{\boldsymbol{X}} = \{i \in [n] \mid \mathcal{E}_i(\boldsymbol{X}) \geq \lambda\}. \tag{5}$$

**Statistical test for diffusion models**   In order to quantify the statistical significance of the anomalous regions detected by using the diffusion model, we consider the concrete example of two-sample test. Note that our method can be extended to other statistical tests using various statistics. In the two-sample test, we compare the test image and the randomly chosen reference image in the anomalous region. Let us define an $n$-dimensional reference input vector,

$$\boldsymbol{X}^{\mathrm{ref}} = (X_1^{\mathrm{ref}}, X_2^{\mathrm{ref}}, \ldots, X_n^{\mathrm{ref}})^\top = \boldsymbol{\mu}^{\mathrm{ref}} + \boldsymbol{\varepsilon}^{\mathrm{ref}}, \quad \text{with } \boldsymbol{\varepsilon}^{\mathrm{ref}} \sim \mathcal{N}(\boldsymbol{0}, \Sigma),$$

where $\boldsymbol{\mu}^{\mathrm{ref}} \in \mathbb{R}^n$ is the unknown true signal vector of the reference image, $\boldsymbol{\varepsilon}^{\mathrm{ref}} \in \mathbb{R}^n$ is the noise component, and $\boldsymbol{X}^{\mathrm{ref}}$ is assumed to be independent of $\boldsymbol{X}$. Then, we consider the following null and alternative hypotheses:

$$\mathrm{H}_0 \colon \frac{1}{|\mathcal{M}_{\boldsymbol{x}}|} \sum_{i \in \mathcal{M}_{\boldsymbol{x}}} \mu_i = \frac{1}{|\mathcal{M}_{\boldsymbol{x}}|} \sum_{i \in \mathcal{M}_{\boldsymbol{x}}} \mu_i^{\mathrm{ref}} \quad \text{vs.} \quad \mathrm{H}_1 \colon \frac{1}{|\mathcal{M}_{\boldsymbol{x}}|} \sum_{i \in \mathcal{M}_{\boldsymbol{x}}} \mu_i \neq \frac{1}{|\mathcal{M}_{\boldsymbol{x}}|} \sum_{i \in \mathcal{M}_{\boldsymbol{x}}} \mu_i^{\mathrm{ref}}, \tag{6}$$

where $\mathrm{H}_0$ is the null hypothesis that the mean pixel values are the same between the test image and the reference image in the observed anomalous region $\mathcal{M}_{\boldsymbol{x}}$, while $\mathrm{H}_1$ is the alternative hypothesis that they are different. A reasonable test statistic for the statistical test in (6) is the difference in mean pixel values between the test image and the reference image in the anomalous region $\mathcal{M}_{\boldsymbol{x}}$, i.e.,

$$T(\boldsymbol{X}, \boldsymbol{X}^{\mathrm{ref}}) = \frac{1}{|\mathcal{M}_{\boldsymbol{x}}|} \sum_{i \in \mathcal{M}_{\boldsymbol{x}}} X_i - \frac{1}{|\mathcal{M}_{\boldsymbol{x}}|} \sum_{i \in \mathcal{M}_{\boldsymbol{x}}} X_i^{\mathrm{ref}} = \boldsymbol{\nu}^\top \begin{pmatrix} \boldsymbol{X} \\ \boldsymbol{X}^{\mathrm{ref}} \end{pmatrix}, \tag{7}$$

where $\boldsymbol{\nu} \in \mathbb{R}^{2n}$ denotes a vector that depends on the anomalous region $\mathcal{M}_{\boldsymbol{x}}$, and hence on $\boldsymbol{x}$ itself, and is defined as

$$\boldsymbol{\nu} = \frac{1}{|\mathcal{M}_{\boldsymbol{x}}|} \begin{pmatrix} \mathbf{1}_{\mathcal{M}_{\boldsymbol{x}}}^n \\ -\mathbf{1}_{\mathcal{M}_{\boldsymbol{x}}}^n \end{pmatrix} \in \mathbb{R}^{2n},$$

where $\mathbf{1}_{\mathcal{C}}^n \in \mathbb{R}^n$ is an $n$-dimensional vector whose elements are 1 if they belong to the set $\mathcal{C}$ and 0 otherwise. In this case, the $p$-value is called the naive $p$-value, and is defined as

$$p_{\text{naive}} = \mathbb{P}_{H_0} \left( |T(\boldsymbol{X}, \boldsymbol{X}^{\text{ref}})| \geq |T(\boldsymbol{x}, \boldsymbol{x}^{\text{ref}})| \right). \tag{8}$$

If we can identify the sampling distribution of the test statistic $T(\boldsymbol{X}, \boldsymbol{X}^{\text{ref}})$, we can compute a valid $p$-value that controls the false positive detection rate (i.e., the type I error rate).

## 4 Selective inference for diffusion-based anomaly localization

In this section, we introduce the SI framework for testing the anomalous regions detected by a diffusion model and propose a method to perform valid hypothesis tests.

### 4.1 Computing valid $p$-values via selective inference

**Complexity of sampling distribution**   As mentioned in §3, we need to identify the sampling distribution of the test statistic $T(\boldsymbol{X}, \boldsymbol{X}^{\text{ref}})$ to compute the $p$-values. If we ignore the fact that the anomalous region is identified by a diffusion model, the test statistic in (7) is simply a linear transformation of the Gaussian random vectors $\boldsymbol{X}$ and $\boldsymbol{X}^{\text{ref}}$, and hence itself follows a Gaussian distribution under the null hypothesis $H_0$:

$$T(\boldsymbol{X}, \boldsymbol{X}^{\text{ref}}) \sim \mathcal{N}(0, \boldsymbol{\nu}^\top \tilde{\Sigma} \boldsymbol{\nu}), \;\; \text{with} \;\; \tilde{\Sigma} = \begin{pmatrix} \Sigma & O_n \\ O_n & \Sigma \end{pmatrix},$$

where $\boldsymbol{X}$ and $\boldsymbol{X}^{\text{ref}}$ are assumed to be independent. However, as mentioned in §3, in reality the dependence of $\boldsymbol{\nu}$ on $\boldsymbol{x}$ via the diffusion model is more intricate, making the sampling distribution intractably complex. Consequently, obtaining this sampling distribution directly is challenging.

**Selective $p$-value via conditional sampling distribution**   Then, we consider the sampling distribution of the test statistic conditional on the event that the anomalous region $\mathcal{M}_{\boldsymbol{X}}$ is the same as the observed anomalous region $\mathcal{M}_{\boldsymbol{x}}$, i.e.,

$$T(\boldsymbol{X}, \boldsymbol{X}^{\text{ref}}) \,|\, \{\mathcal{M}_{\boldsymbol{X}} = \mathcal{M}_{\boldsymbol{x}}\}.$$

In the context of SI, to make the characterization of the conditional sampling distribution manageable, we also incorporate conditioning on the nuisance parameter that is independent of the test statistic. As a result, the calculation of the conditional sampling distribution in SI can be reduced to a one-dimensional search problem in an $n$-dimensional data space. The sufficient statistic of the nuisance parameter $\mathcal{Q}_{\boldsymbol{X}, \boldsymbol{X}^{\text{ref}}}$ is written as

$$\mathcal{Q}_{\boldsymbol{X}, \boldsymbol{X}^{\text{ref}}} = \left( I_{2n} - \frac{\tilde{\Sigma} \boldsymbol{\nu} \boldsymbol{\nu}^\top}{\boldsymbol{\nu}^\top \tilde{\Sigma} \boldsymbol{\nu}} \right) \begin{pmatrix} \boldsymbol{X} \\ \boldsymbol{X}^{\text{ref}} \end{pmatrix},$$

where $I_{2n}$ is the identity matrix of size $2n$. By additional conditioning on the nuisance parameter, the selective $p$-value is defined as

$$p_{\text{selective}} = \mathbb{P}_{H_0} \left( |T(\boldsymbol{X}, \boldsymbol{X}^{\text{ref}})| \geq |T(\boldsymbol{x}, \boldsymbol{x}^{\text{ref}})| \,\big|\, \mathcal{M}_{\boldsymbol{X}} = \mathcal{M}_{\boldsymbol{x}}, \mathcal{Q}_{\boldsymbol{X}, \boldsymbol{X}^{\text{ref}}} = \mathcal{Q}_{\boldsymbol{x}, \boldsymbol{x}^{\text{ref}}} \right). \tag{9}$$

The following theorem establishes that the selective $p$-value is a valid $p$-value for controlling the false positive detection rate for any significance level $\alpha \in (0, 1)$.

**Theorem 4.1.** *The selective $p$-value in* (9) *is valid for controlling the false positive detection rate, i.e,*

$$\mathbb{P}_{H_0} \left( p_{\text{selective}} \leq \alpha \,\big|\, \mathcal{M}_{\boldsymbol{X}} = \mathcal{M}_{\boldsymbol{x}}, \mathcal{Q}_{\boldsymbol{X}, \boldsymbol{X}^{\text{ref}}} = \mathcal{Q}_{\boldsymbol{x}, \boldsymbol{x}^{\text{ref}}} \right) = \alpha, \; \forall \alpha \in (0, 1).$$

*Then, the selective $p$-value satisfies the following condition:*

$$\mathbb{P}_{H_0}(p_{\text{selective}} \leq \alpha) = \alpha, \; \forall \alpha \in (0, 1).$$

The proof of Theorem 4.1 is given in Appendix A.1. The following theorem states that the selective $p$-value can be analytically derived from the conditional sampling distribution, which follows a truncated Gaussian distribution.

**Theorem 4.2.** *Consider the truncation intervals defined as*

$$\mathcal{Z} = \left\{ z \in \mathbb{R} \mid \mathcal{M}_{\boldsymbol{X}(z)} = \mathcal{M}_{\boldsymbol{x}} \right\}, \tag{10}$$

*where $\boldsymbol{X}(z)$ is defined as*

$$\boldsymbol{X}(z) = \boldsymbol{a}_{1:n} + \boldsymbol{b}_{1:n} z, \ \ where \ \ \boldsymbol{a} = \mathcal{Q}_{\boldsymbol{x}}, \ \boldsymbol{b} = \frac{\tilde{\Sigma}\boldsymbol{\nu}}{\boldsymbol{\nu}^\top \tilde{\Sigma} \boldsymbol{\nu}}, \tag{11}$$

*and $\boldsymbol{a}_{1:n}$ and $\boldsymbol{b}_{1:n}$ denote the first $n$ elements of $\boldsymbol{a}, \boldsymbol{b} \in \mathbb{R}^{2n}$, respectively. Then, the selective $p$-value in (9) can be rewritten as*

$$p_{\text{selective}} = \mathbb{P}_{\text{H}_0} \left( |T(\boldsymbol{X}(Z), \boldsymbol{X}^{\text{ref}}(Z))| \geq |T(\boldsymbol{x}, \boldsymbol{x}^{\text{ref}})| \mid Z \in \mathcal{Z} \right). \tag{12}$$

*The conditional sampling distribution of the test statistic $T(\boldsymbol{X}(Z), \boldsymbol{X}^{\text{ref}}(Z)) \mid \{Z \in \mathcal{Z}\}$ follows a truncated Gaussian distribution $\mathcal{TN}(0, \boldsymbol{\nu}^\top \tilde{\Sigma} \boldsymbol{\nu})$.*

The proof of Theorem 4.2 is given in Appendix A.2. Once the truncation intervals $\mathcal{Z}$ are identified, computing the selective $p$-value in (12) becomes straightforward. Therefore, the remaining task is the identification of $\mathcal{Z}$.

### 4.2  Identification of truncation intervals $\mathcal{Z}$

**Identification of subinterval $\mathcal{Z}^{\text{sub}}$**  To tackle the identification of the truncation intervals $\mathcal{Z}$, we employ a divide-and-conquer strategy. Directly characterizing $\mathcal{Z}$ is challenging due to the complexity of the diffusion model's computation. Our method decomposes the $n$-dimensional data space into a collection of polyhedra by imposing additional conditioning, a process we refer to as *over-conditioning (OC)* (Duy & Takeuchi, 2022). Each polyhedron in the $n$-dimensional space corresponds to an interval in the one-dimensional space $\mathcal{Z}$. Thus, we can examine these intervals sequentially in $\mathcal{Z}$ to determine whether they yield the same selected anomalous regions as observed. To this end, we need to identify a subinterval $\mathcal{Z}^{\text{sub}} \subseteq \mathcal{Z}$. We show that the anomalous region from a diffusion model can be characterized by a piecewise-linear mapping. Exploiting this property, the subinterval $\mathcal{Z}^{\text{sub}}(\boldsymbol{a} + \boldsymbol{b}z)$ can be computed analytically for each $z \in \mathbb{R}$ by solving a system of linear inequalities (see Appendix B). This piecewise-linearity holds for U-Nets composed of piecewise-linear components such as ReLU activations and pooling layers; U-Nets with smooth activation functions (e.g., SiLU) can also be handled by replacing each activation with a piecewise-linear approximation at inference time (see Appendix C).

**Identification of $\mathcal{Z}$ via parametric programming**  Over-conditioning causes a reduction in power due to excessive conditioning. A technique called *parametric programming* is utilized to explore all intervals along the one-dimensional line, resulting in (10). The truncation intervals $\mathcal{Z}$ can be represented using $\mathcal{Z}^{\text{sub}}$ as

$$\mathcal{Z} = \bigcup_{z \in \mathbb{R} \mid \mathcal{M}_{\boldsymbol{X}(z)} = \mathcal{M}_{\boldsymbol{x}}} \mathcal{Z}^{\text{sub}}(\boldsymbol{a} + \boldsymbol{b}z).$$

Figure 2 illustrates the conditional sampling distribution determined by these truncation intervals. An algorithm for computing the selective $p$-value is summarized in Algorithm 2. In our implementation, the search in Algorithm 2 is restricted to the range $|z| \leq 10\sigma + |T(\boldsymbol{x}, \boldsymbol{x}^{\text{ref}})|$, i.e., the range extending $10\sigma$ beyond the absolute value of the observed test statistic, where $\sigma = \sqrt{\boldsymbol{\nu}^\top \tilde{\Sigma} \boldsymbol{\nu}}$ is the standard deviation of the test statistic; the probability mass of the Gaussian distribution outside this range is less than $2 \times 10^{-23}$, and hence the effect of this restriction on the selective $p$-value is numerically negligible. In addition, our implementation adopts the method of Shiraishi et al. (2024a), which keeps track of the lower and upper bounds of the selective $p$-value during the search and enables early termination of the search once the $p$-value is evaluated

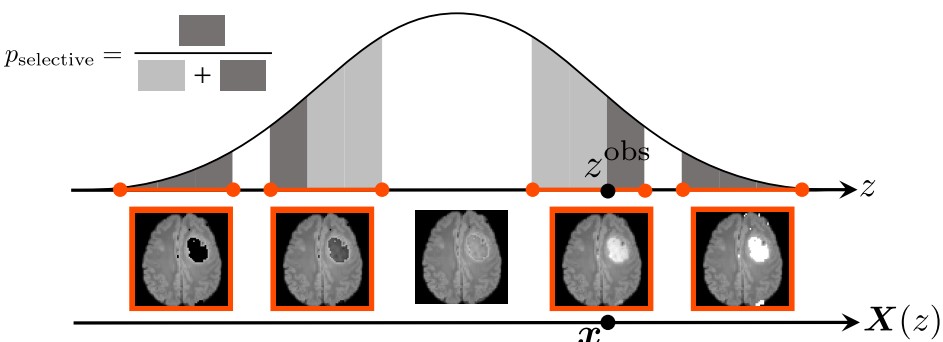

Figure 2: Schematic illustration of the selective inference procedure for diffusion models. It shows how the image $\boldsymbol{X}(z)$ changes with $z$. The truncation intervals that yield the same anomalous region $\mathcal{M}_{\boldsymbol{X}(z)}$ as the observed anomalous region $\mathcal{M}_{\boldsymbol{x}}$ define the conditional sampling distribution. The $p_{\text{selective}}$ denotes the proportion of probability mass within the truncation intervals.

with sufficient precision, without compromising the validity. The computational complexity of Algorithm 2 is analyzed in Appendix D.

---

**Algorithm 2** Selective $p$-value by Parametric Programming

---

**Require:** $\boldsymbol{x}, \boldsymbol{x}^{\text{ref}}$
1: Initialize $\mathcal{Z} \leftarrow \emptyset$
2: Compute $\mathcal{M}_{\boldsymbol{x}}, \boldsymbol{a}, \boldsymbol{b}$, and $T(\boldsymbol{x}, \boldsymbol{x}^{\text{ref}})$ by (5), (7), (11)
3: Initialize $z \leftarrow -(10\sigma + |T(\boldsymbol{x}, \boldsymbol{x}^{\text{ref}})|)$
4: **while** $z \leq 10\sigma + |T(\boldsymbol{x}, \boldsymbol{x}^{\text{ref}})|$ **do**
5:     Compute $\mathcal{Z}^{\text{sub}}(\boldsymbol{a} + \boldsymbol{b}z)$ and $\mathcal{M}_{\boldsymbol{X}(z)}$ by (13)
6:     **if** $\mathcal{M}_{\boldsymbol{X}(z)} = \mathcal{M}_{\boldsymbol{x}}$ **then**
7:         $\mathcal{Z} \leftarrow \mathcal{Z} \cup \mathcal{Z}^{\text{sub}}(\boldsymbol{a} + \boldsymbol{b}z)$
8:     **end if**
9:     $z \leftarrow \max \mathcal{Z}^{\text{sub}}(\boldsymbol{a} + \boldsymbol{b}z) + \gamma$, where $0 < \gamma \ll 1$
10: **end while**
11: $p_{\text{selective}} = \mathbb{P}_{\text{H}_0} \left( |T(\boldsymbol{X}(Z), \boldsymbol{X}^{\text{ref}}(Z))| \geq |T(\boldsymbol{x}, \boldsymbol{x}^{\text{ref}})| \mid Z \in \mathcal{Z} \right)$
**Ensure:** $p_{\text{selective}}$

---

## 5 Experiments

We compared our proposed method (`proposed`) on type I error rate and power with the following methods (see Appendix E for details of those methods):

- `w/o-pp`: An ablation study without the parametric programming technique described in §4.2. The $p$-value is computed using (12), with $\mathcal{Z}$ replaced by $\mathcal{Z}^{\text{sub}}(\boldsymbol{a} + \boldsymbol{b}z^{\text{obs}})$.

- `naive`: This method is conventionally used in practice. The $p$-value is computed as (8).

- `bonferroni`: This method uses the Bonferroni correction. Bonferroni correction is widely used for multiple testing correction.

- `permutation`: This method uses the permutation test. A permutation test is widely used for non-parametric hypothesis testing.

To the best of our knowledge, these methods cover all practically relevant alternatives for statistical inference on regions selected by a diffusion model. The `naive` and `permutation` approaches are commonly adopted

in practice, although neither is guaranteed to control the type I error rate under data-dependent region selection. Among multiple-testing procedures, the Bonferroni correction is essentially the only practical baseline, since it requires only the number of hypotheses, whereas more powerful procedures such as Holm's or Hochberg's method require the computation of $p$-values for all hypotheses. In our setting, this would require evaluating all possible image regions, which can grow exponentially with the number of pixels, and is computationally infeasible. Finally, `w/o-pp` is included as the standard selective inference formulation to isolate the contribution of our proposed parametric programming technique. The architecture of diffusion models used in all experiments is detailed in Appendix F. The computational complexity and runtime analysis of the proposed method are presented in Appendix D, and all the hyperparameters are consolidated in Appendix G.1. We executed all experiments on AMD EPYC 9474F processor, 48-core 3.6GHz CPU and 768GB memory.

## 5.1 Numerical experiments

**Experimental setup** Experiments on the type I error rate and power were conducted with two types of covariance matrices: independent $\Sigma = I_n \in \mathbb{R}^{n \times n}$ and correlation $\Sigma = (0.5^{|i-j|})_{ij} \in \mathbb{R}^{n \times n}$. In the type I error rate experiments, we used only normal images. The synthetic dataset for normal images is generated to follow $\boldsymbol{X} \sim \mathcal{N}(\boldsymbol{0}, \Sigma)$. We generated 1,000 normal images for $n \in \{64, 256, 1024, 4096\}$. In the power experiments, we used only abnormal images. We generated 1,000 abnormal images $\boldsymbol{X} \sim \mathcal{N}(\boldsymbol{\mu}, \Sigma)$. The mean vector $\boldsymbol{\mu}$ is defined as $\mu_i = \Delta$ for all $i \in \mathcal{S}$, and $\mu_i = 0$ for all $i \in [n] \backslash \mathcal{S}$, where $\mathcal{S} \subset [n]$ is the anomalous region with its position randomly chosen. The image size of the abnormal images was set to 4096, with signals $\Delta \in \{1, 2, 3, 4\}$. In all experiments, we generated the synthetic dataset for 1,000 reference images to follow $\boldsymbol{X}^{\mathrm{ref}} \sim \mathcal{N}(\boldsymbol{0}, \Sigma)$. The threshold was set to $\lambda = 0.8$, and the kernel size of the averaging filter was set to 3. All experiments were conducted under the significance level $\alpha = 0.05$. The diffusion models were trained on the normal images from the synthetic dataset. The diffusion models were trained with $T = 1000$ and the initial time step of the reverse process was set to $T' = 460$, and the reconstruction was performed using 5-step sampling (see Appendix H for details). The noise schedule $\beta_1, \beta_2, \ldots, \beta_T$ was set to linear. In all experiments, we generated normal-looking images through probabilistic sampling, with $\eta$ set to 1. In addition, we conducted robustness experiments against non-Gaussian noise; the details are described in Appendix I.

**Results** Figure 3 shows the comparison results of type I error rates. The proposed methods `proposed` and `w/o-pp` can control the type I error rate at the significance level $\alpha$, and `bonferroni` can control the type I error rate below the significance level $\alpha$. In contrast, `naive` and `permutation` cannot control the type I error rate in general. The type I error rate of `naive` far exceeds the significance level in both the independence and correlation settings. Although `permutation` keeps the type I error rate around the significance level in the independence setting, where the pixel values are exchangeable under the null hypothesis, its type I error rate far exceeds the significance level in the correlation setting, where the exchangeability assumption is violated. Hence, the detection results of these two methods are not statistically reliable. Figure 4 shows the comparison results of powers. Since `naive` and `permutation` cannot control the type I error rate, their powers are not considered. Among the methods that can control the type I error rate, the `proposed` has the highest power. The power of the `proposed` increases with the signal strength $\Delta$ in both the independence and correlation settings. The ablation study `w/o-pp` is over-conditioned and `bonferroni` is conservative because there are many hypotheses, so they have low power. In particular, the comparison between `proposed` and `w/o-pp` serves as an ablation study that directly validates the effectiveness of Algorithm 2: both variants control the type I error rate, but exploring all the intervals via parametric programming removes the excessive conditioning and thereby substantially improves the power.

## 5.2 Real-world data experiments

**Experimental setup** We conducted experiments using T2-FLAIR MRI brain scans from the Brain Tumor Segmentation (BraTS) 2023 dataset (Karargyris et al., 2023; LaBella et al., 2023) and MVTec AD dataset (Bergmann et al., 2019). The details of the experimental settings are described in Appendix G.

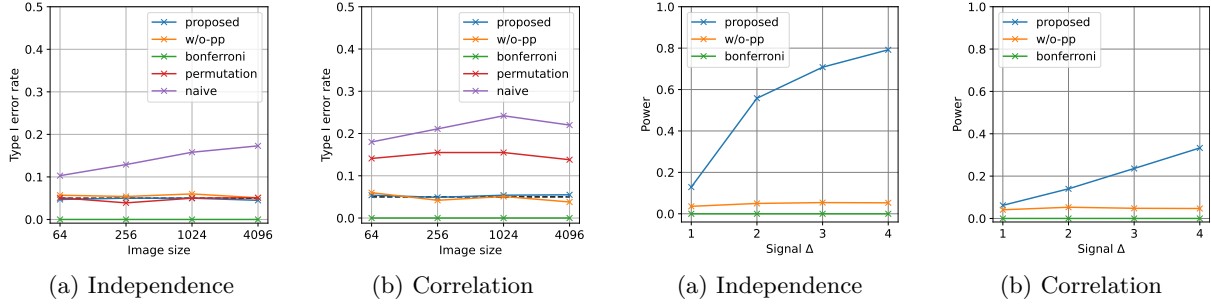

(a) Independence  (b) Correlation  (a) Independence  (b) Correlation

Figure 3: Type I error rate comparison        Figure 4: Power comparison

Table 1: Type I error rate and power comparison on real-world datasets at the significance level of $\alpha(= 0.05)$. The `proposed` method yields empirical type I error rates close to the nominal level $\alpha$ across all datasets and achieves higher power than `bonferroni`, which controls the type I error rate below $\alpha$, whereas `naive` fails to control the type I error rate and is therefore unreliable.

|  | naive | | bonferroni | | proposed | |
|---|---|---|---|---|---|---|
| Dataset | Type I Error | Power | Type I Error | Power | Type I Error | Power |
| Bottle | 0.46 | N.A | 0.00 | 0.00 | **0.04** | **0.18** |
| Cable | 0.88 | N.A | 0.00 | 0.00 | **0.02** | **0.40** |
| Grid | 0.82 | N.A | 0.00 | 0.00 | **0.06** | **0.34** |
| Transistor | 0.86 | N.A | 0.00 | 0.00 | **0.08** | **0.28** |
| BraTS (T2-FLAIR) | 0.59 | N.A | 0.00 | 0.00 | **0.05** | **0.28** |

**Results**   Table 1 shows the comparison of the type I error rate and power. The `naive` cannot control the type I error rate, while the `proposed` method yields empirical type I error rates close to the significance level $\alpha$ and `bonferroni` controls the type I error rate below $\alpha$. The `proposed` has higher power than `bonferroni`. Specifically, the type I error rates of `naive` range from 0.46 to 0.88 across the datasets, far above the significance level $\alpha = 0.05$. `bonferroni` also controls the type I error rate, but its power is 0.00 on all datasets due to its conservativeness, while the `proposed` method attains powers of 0.18–0.40. Figure 5 shows representative results on the BraTS dataset and four MVTec AD categories, comparing the naive $p$-value and the proposed selective $p$-value on a normal image (left) and an anomalous image (right) for each dataset. For normal images, the diffusion model still reports a detected region and the naive $p$-value is small (e.g., $p_{\text{naive}} = 0.000$ for the normal bottle image), leading to a false positive, whereas the selective $p$-value is large (e.g., $p_{\text{selective}} = 0.907$), correctly indicating that the detected region is not statistically significant. For anomalous images, both $p$-values are small, so the selective $p$-value retains the ability to flag true anomalies (e.g., $p_{\text{selective}} = 0.010$ for the defective bottle). Thus, the naive $p$-value cannot distinguish spurious detections on normal images from genuine anomalies, whereas the selective $p$-value can.

## 6   Conclusions

We introduced the DAL-Test, a novel statistical procedure for anomaly localization identified by a diffusion model. With the proposed DAL-Test, the false positive detection rate can be controlled at the significance level because statistical inference is conducted conditional on the fact that the anomalous regions are identified by using a diffusion model. We demonstrated that the DAL-Test has higher power than the Bonferroni correction, which is the only computationally feasible valid baseline in our setting. We also empirically confirmed that the DAL-Test is robust against moderate deviations from the Gaussian noise assumption, and that its computational cost grows far more slowly than the exponential worst-case order in practice. An important direction for future work is to extend the proposed framework to diffusion architectures with multivariate nonlinearities such as layer normalization and softmax attention (e.g., ViT-based diffusion models) by integrating the SI techniques recently developed for Vision Transformers (Shiraishi et al., 2024b).

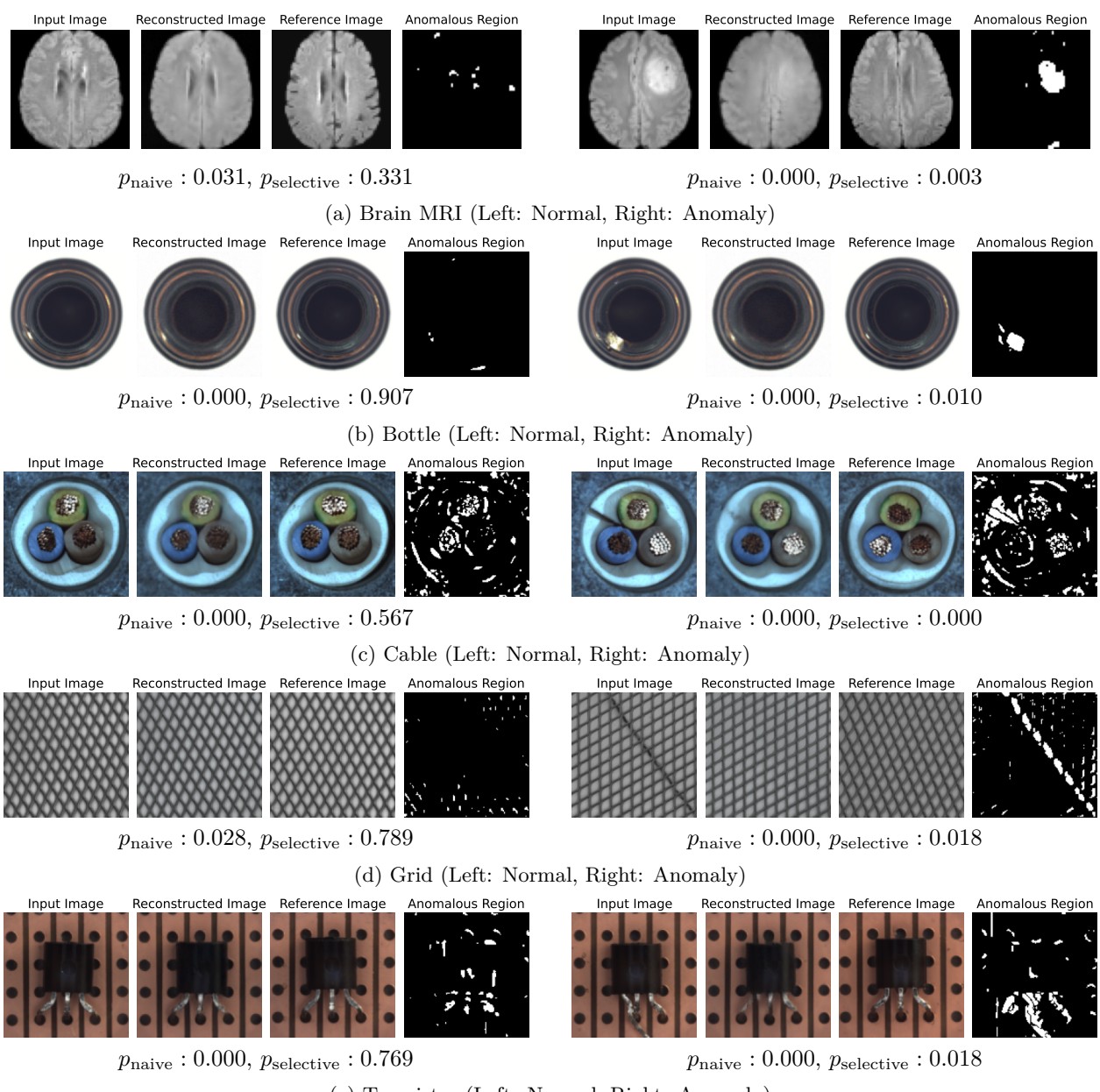

Figure 5: Results of the `proposed` and `naive` methods on the BraTS dataset (one example) and the MVTec AD dataset (four categories). For each dataset, the left image is a normal sample and the right image is an anomalous sample. The $p_{\text{selective}}$ is high for normal images (true negative) and low for anomalous images (true positive), demonstrating that the proposed method correctly controls the false positive detection rate. In contrast, the $p_{\text{naive}}$ remains low for both normal and anomalous images, indicating an inflated false positive rate.

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

# A Proofs

## A.1 Proof of Theorem 4.1

Under the null hypothesis, the probability integral transform implies that

$$p_{\text{selective}} \mid \left\{ \mathcal{M}_{\boldsymbol{X}} = \mathcal{M}_{\boldsymbol{x}}, \mathcal{Q}_{\boldsymbol{X}, \boldsymbol{X}^{\text{ref}}} = \mathcal{Q}_{\boldsymbol{x}, \boldsymbol{x}^{\text{ref}}} \right\} \sim \text{Uniform}(0, 1),$$

and hence for any $\alpha \in (0, 1)$,

$$\mathbb{P} \left( p_{\text{selective}} \leq \alpha \mid \left\{ \mathcal{M}_{\boldsymbol{X}} = \mathcal{M}_{\boldsymbol{x}}, \mathcal{Q}_{\boldsymbol{X}, \boldsymbol{X}^{\text{ref}}} = \mathcal{Q}_{\boldsymbol{x}, \boldsymbol{x}^{\text{ref}}} \right\} \right) = \alpha, \ \forall \alpha \in (0, 1).$$

By marginalizing over the nuisance parameter $\mathcal{Q}_{\boldsymbol{x}}$, we have

$$\mathbb{P} \left( p_{\text{selective}} \leq \alpha \mid \mathcal{M}_{\boldsymbol{X}} = \mathcal{M}_{\boldsymbol{x}}, \mathcal{Q}_{\boldsymbol{X}, \boldsymbol{X}^{\text{ref}}} = \mathcal{Q}_{\boldsymbol{x}, \boldsymbol{x}^{\text{ref}}} \right)$$

$$= \int_{\mathbb{R}^n} \mathbb{P}_{\text{H}_0}(p_{\text{selective}} \leq \alpha \mid \mathcal{M}_{\boldsymbol{X}} = \mathcal{M}_{\boldsymbol{x}}, \mathcal{Q}_{\boldsymbol{X}, \boldsymbol{X}^{\text{ref}}} = \mathcal{Q}_{\boldsymbol{x}, \boldsymbol{x}^{\text{ref}}}) \, \mathbb{P}_{\text{H}_0}(\mathcal{Q}_{\boldsymbol{X}, \boldsymbol{X}^{\text{ref}}} = \mathcal{Q}_{\boldsymbol{x}, \boldsymbol{x}^{\text{ref}}} \mid \mathcal{M}_{\boldsymbol{X}} = \mathcal{M}_{\boldsymbol{x}}) d\mathcal{Q}_{\boldsymbol{x}}$$

$$= \alpha \int_{\mathbb{R}^n} \mathbb{P}_{\text{H}_0}(\mathcal{Q}_{\boldsymbol{X}, \boldsymbol{X}^{\text{ref}}} = \mathcal{Q}_{\boldsymbol{x}, \boldsymbol{x}^{\text{ref}}} \mid \mathcal{M}_{\boldsymbol{X}} = \mathcal{M}_{\boldsymbol{x}}) d\mathcal{Q}_{\boldsymbol{x}}.$$

$$= \alpha$$

Therefore, we have

$$\mathbb{P}_{\text{H}_0} \left( p_{\text{selective}} \leq \alpha \right)$$

$$= \sum_{\mathcal{M}_{\boldsymbol{x}} \in 2^{[n]}} \mathbb{P}_{\text{H}_0}(\mathcal{M}_{\boldsymbol{x}}) \, \mathbb{P}_{\text{H}_0}(p_{\text{selective}} \leq \alpha \mid \mathcal{M}_{\boldsymbol{X}} = \mathcal{M}_{\boldsymbol{x}})$$

$$= \alpha \sum_{\mathcal{M}_{\boldsymbol{x}} \in 2^{[n]}} \mathbb{P}_{\text{H}_0}(\mathcal{M}_{\boldsymbol{x}})$$

$$= \alpha$$

## A.2 Proof of Theorem 4.2

The conditioning on $\mathcal{Q}_{\boldsymbol{X}, \boldsymbol{X}^{\text{ref}}} = \mathcal{Q}_{\boldsymbol{x}, \boldsymbol{x}^{\text{ref}}}$ implies

$$\mathcal{Q}_{\boldsymbol{X}, \boldsymbol{X}^{\text{ref}}} = \mathcal{Q}_{\boldsymbol{x}, \boldsymbol{x}^{\text{ref}}} \Leftrightarrow \left( I_{2n} - \frac{\tilde{\Sigma} \boldsymbol{\nu} \boldsymbol{\nu}^{\top}}{\boldsymbol{\nu}^{\top} \tilde{\Sigma} \boldsymbol{\nu}} \right) \begin{pmatrix} \boldsymbol{X} \\ \boldsymbol{X}^{\text{ref}} \end{pmatrix} = \mathcal{Q}_{\boldsymbol{x}, \boldsymbol{x}^{\text{ref}}} \Leftrightarrow \begin{pmatrix} \boldsymbol{X} \\ \boldsymbol{X}^{\text{ref}} \end{pmatrix} = \boldsymbol{a} + \boldsymbol{b} z,$$

where $z = T(\boldsymbol{X}, \boldsymbol{X}^{\text{ref}}) \in \mathbb{R}$. Hence,

$$\left\{ \begin{pmatrix} \boldsymbol{X} \\ \boldsymbol{X}^{\text{ref}} \end{pmatrix} \Bigg| \mathcal{M}_{\boldsymbol{X}} = \mathcal{M}_{\boldsymbol{x}}, \mathcal{Q}_{\boldsymbol{X}, \boldsymbol{X}^{\text{ref}}} = \mathcal{Q}_{\boldsymbol{x}, \boldsymbol{x}^{\text{ref}}} \right\}$$

$$= \left\{ \begin{pmatrix} \boldsymbol{X} \\ \boldsymbol{X}^{\text{ref}} \end{pmatrix} \Bigg| \mathcal{M}_{\boldsymbol{X}} = \mathcal{M}_{\boldsymbol{x}}, \begin{pmatrix} \boldsymbol{X} \\ \boldsymbol{X}^{\text{ref}} \end{pmatrix} = \boldsymbol{a} + \boldsymbol{b} z \right\}$$

$$= \left\{ \begin{pmatrix} \boldsymbol{X} \\ \boldsymbol{X}^{\text{ref}} \end{pmatrix} \Bigg| \mathcal{M}_{\boldsymbol{X}(z)} = \mathcal{M}_{\boldsymbol{x}} \right\}$$

$$= \left\{ \boldsymbol{a} + \boldsymbol{b} z \mid z \in \mathcal{Z} \right\},$$

where $\boldsymbol{X}(z) = \boldsymbol{a}_{1:n} + \boldsymbol{b}_{1:n} z$. As a result,

$$T(\boldsymbol{X}, \boldsymbol{X}^{\text{ref}}) \mid \left\{ \mathcal{M}_{\boldsymbol{X}} = \mathcal{M}_{\boldsymbol{x}}, \mathcal{Q}_{\boldsymbol{X}, \boldsymbol{X}^{\text{ref}}} = \mathcal{Q}_{\boldsymbol{x}, \boldsymbol{x}^{\text{ref}}} \right\} \sim \mathcal{TN}(0, \boldsymbol{\nu}^{\top} \tilde{\Sigma} \boldsymbol{\nu})$$

# B   Calculating the subinterval $\mathcal{Z}^{\text{sub}}$ for diffusion models

We show that a reconstruction error $\mathcal{E}$ via diffusion models can be expressed as a piecewise-linear function of $\boldsymbol{X}$. To show this, we see that both the forward process and reverse process of the diffusion model are piecewise-linear functions as long as we employ a class of U-Net described below. It is easy to see the piecewise-linearity of the forward process as long as we fix the random seed for $\epsilon_t$. To make the reverse process a piecewise-linear function, we employ a U-Net composed of piecewise-linear components such as ReLU activation functions and pooling layers. Then, $\epsilon_\theta^{(t)}(\mathbf{x}_t)$ is represented as a piecewise-linear function of $\mathbf{x}_t$. Moreover, since $f_\theta^{(t)}(\mathbf{x}_t)$ in (3) is a composite function of $\epsilon_\theta^{(t)}(\mathbf{x}_t)$, it is also a piecewise-linear function. By combining them together, we see that $\mathbf{x}_{t-1}$ is written as a piecewise-linear function of $\mathbf{x}_t$. Therefore, the entire reconstruction process is a piecewise-linear function since it just repeats the above operation multiple times (see Algorithm 1). As a result, the entire mapping $\mathcal{D}(\boldsymbol{X})$ of the diffusion model is a piecewise-linear function of the input image $\boldsymbol{X}$. Moreover, since the averaging filter $\mathcal{F}$ and the absolute operation are also piecewise-linear functions, $|\mathcal{F}(\boldsymbol{X} - \mathcal{D}(\boldsymbol{X}))|(= \mathcal{E}(\boldsymbol{X}))$ is piecewise-linear. Exploiting this piecewise-linearity, the interval $\mathcal{Z}^{\text{sub}}$ can be computed. The following theorem states that the subinterval $\mathcal{Z}^{\text{sub}}(\boldsymbol{a} + \boldsymbol{b}z)$ can be computed by solving a set of linear inequalities.

**Theorem B.1.** *The piecewise-linear mapping $\mathcal{A}(\boldsymbol{X})$ can be expressed as a linear function of the input image $\boldsymbol{X}$ on each polyhedral region $\mathcal{P}_k$.*

$$\forall \boldsymbol{X} \in \mathcal{P}^{(k)}, \ \mathcal{A}(\boldsymbol{X}) = \boldsymbol{\delta}^{(k)} + \boldsymbol{\Delta}^{(k)}\boldsymbol{X},$$

*where $\boldsymbol{\delta}^{(k)}$ and $\boldsymbol{\Delta}^{(k)}$ for $k \in [K]$ are the constant vector and the coefficient matrix with appropriate dimensions for the $k$-th polyhedron, respectively. Using the notation in (12), since the input image $\boldsymbol{X}(z)$ is restricted on a one-dimensional line, each component of the output of $\mathcal{A}$ is written as*

$$\forall z \in [L_i^{(k')}, U_i^{(k')}], \ \mathcal{A}_i(\boldsymbol{X}(z)) = \kappa_i^{(k')} + \rho_i^{(k')}z,$$

*where $\kappa_i^{(k')} \in \mathbb{R}$ and $\rho_i^{(k')} \in \mathbb{R}$ for $k' \in [K_i']$ are the coefficient and the constant of the $k'$-th interval $[L_i^{(k')}, U_i^{(k')}]$, and $K_i'$ is the number of linear pieces of $\mathcal{A}_i$. For each $i \in [n]$, there exists $k' \in [K_i']$ such that $z \in [L_i^{(k')}, U_i^{(k')}]$, then the inequality $\mathcal{A}_i(\boldsymbol{X}(z)) \geq \lambda$, can be solved as*

$$[L_i(z), U_i(z)] \coloneqq \begin{cases} \left[\max\left(L_i^{(k')}, (\lambda - \rho_i^{(k')})/\kappa_i^{(k')}\right), U_i^{(k')}\right] & \text{if } \kappa_i^{(k')} > 0, \\ \left[L_i^{(k')}, \min\left(U_i^{(k')}, (\lambda - \rho_i^{(k')})/\kappa_i^{(k')}\right)\right] & \text{if } \kappa_i^{(k')} < 0. \end{cases}$$

By applying the above theorem, we denote the subinterval as

$$\mathcal{Z}^{\text{sub}}(\boldsymbol{a} + \boldsymbol{b}z) = \bigcap_{i \in [n]} [L_i(z), U_i(z)]. \tag{13}$$

# C   Application to U-Nets with smooth activation functions

While our main experiments employ a U-Net composed of piecewise-linear components such as ReLU activation functions, most recent pre-trained diffusion models adopt smooth activation functions such as SiLU (Swish). In this appendix, we show that the proposed method can also be applied to such architectures by replacing each smooth scalar activation with a fine piecewise-linear approximation at inference time. This replacement requires only the pre-trained weights and forward evaluations; no retraining and no access to the training data are needed, so it is directly applicable to off-the-shelf pre-trained diffusion models. The approximated network is then used both to detect the anomalous region and to compute the truncation intervals. Therefore, the selective $p$-value is exactly valid for the detector that is actually used, and the approximation does not affect the validity guarantee. Since the approximation error of each scalar activation can be made arbitrarily small by increasing the number of linear segments, the output of the approximated network can be made arbitrarily close to that of the original network over a bounded activation range. Consequently, the

effect on the detected region is expected to be negligible except for pixels whose anomaly scores lie very close to the threshold. We empirically verified this strategy as follows. We trained the same U-Net architecture as in Appendix F with SiLU activations, and replaced each SiLU with a piecewise-linear approximation with 8 knots placed on $[-8, 8]$ so as to equalize the approximation error across segments (de Boor, 1973). Figure 6 shows the SiLU function and its piecewise-linear approximation; the knots are concentrated in the high-curvature region around the origin, and the two functions are visually indistinguishable. We then conducted the same type I error rate experiments as in §5.1, i.e., the independence and correlation settings with image sizes $n \in \{64, 256, 1024, 4096\}$. Figure 7 shows the results. The `proposed` method controls the type I error rate at the significance level $\alpha = 0.05$ in both settings for all image sizes, whereas `naive` fails to do so; the behaviors of all methods are consistent with those in Figure 3.

Finally, we clarify the applicability of the proposed method to the other components of modern diffusion architectures. Batch normalization is covered by our framework without any approximation, because it uses frozen statistics at inference time and is therefore an affine map, which preserves the piecewise-linearity. In contrast, input-dependent normalization layers (e.g., layer normalization and group normalization) and softmax attention modules are multivariate nonlinearities, and are not covered by the approximation of individual scalar activations described above. We note that SI has recently been extended to attention-based region selection in Vision Transformers (Shiraishi et al., 2024b), which indicates that the SI framework itself is not inherently restricted to piecewise-linear networks; integrating such techniques with the recursive reconstruction process of diffusion models is an important direction for future work.

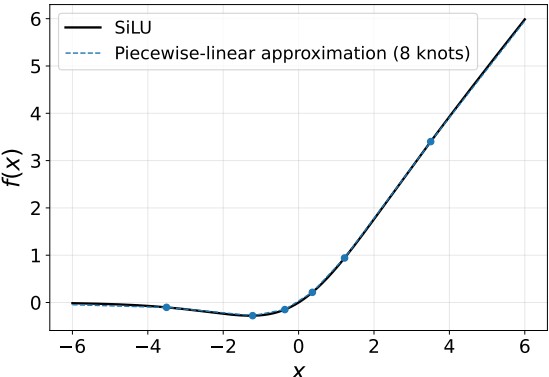

Figure 6: SiLU activation function and its piecewise-linear approximation with 8 knots. The knots (dots) are placed so as to equalize the approximation error across segments, and are therefore concentrated in the high-curvature region around the origin.

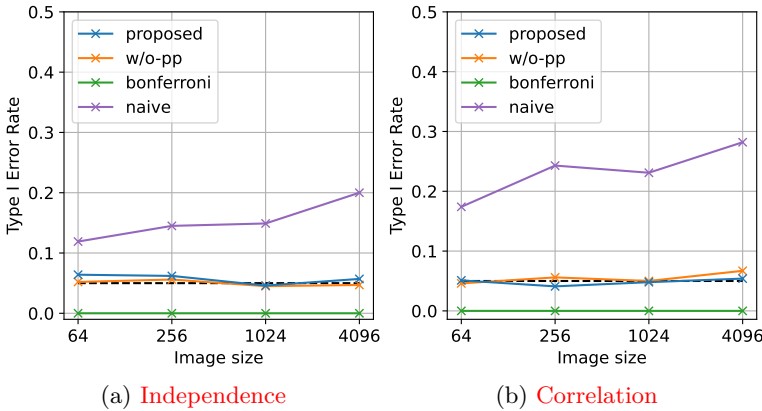

(a) Independence

(b) Correlation

Figure 7: Type I error rate comparison for the U-Net with SiLU activations handled via the piecewise-linear approximation.

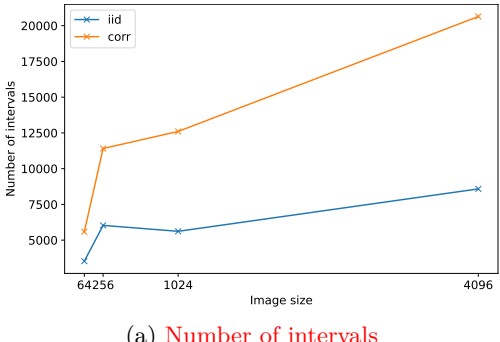 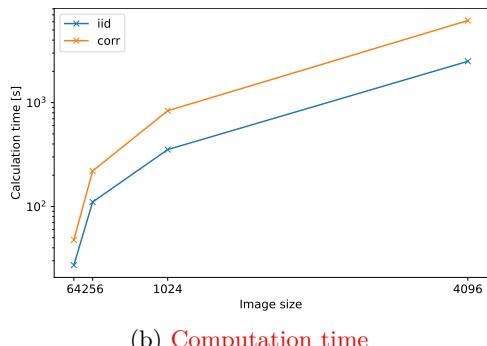

(a) Number of intervals

(b) Computation time

Figure 8: The number of intervals searched by Algorithm 2 (a) and the computation time (b) when changing the image size for the synthetic data. Results are shown for both synthetic data with independent (iid) and correlation (corr) noise.

## D  Computational complexity and runtime analysis

In this appendix, we analyze the computational complexity of Algorithm 2 and empirically evaluate its scalability and runtime.

**Cost of one iteration**  The cost of one iteration of Algorithm 2, i.e., the cost of computing the subinterval $\mathcal{Z}^{\mathrm{sub}}$ in (13), is a small constant multiple of one forward pass of the reconstruction process (Algorithm 1). Restricted to the one-dimensional line $\boldsymbol{X}(z) = \boldsymbol{a} + \boldsymbol{b}z$, every intermediate quantity of the network is a linear function of $z$ within the current linear piece. Hence, it suffices to propagate an intercept and a slope with respect to $z$ for each scalar unit, and each piecewise-linear unit contributes one linear inequality in $z$ that is solved in closed form. The subinterval $\mathcal{Z}^{\mathrm{sub}}$ is then obtained by intersecting the resulting intervals $[L_i(z), U_i(z)]$ defined in Appendix B, i.e., by taking the maximum of the lower bounds and the minimum of the upper bounds. The cost of one iteration is therefore $O(K \cdot C_{\mathrm{U\text{-}Net}})$, where $K$ is the number of denoising steps ($K = 5$ in our numerical experiments) and $C_{\mathrm{U\text{-}Net}}$ is the cost of one U-Net evaluation; that is, it scales linearly with both the number of diffusion steps and the network size.

**Number of iterations**  The total cost of Algorithm 2 is the cost of one iteration multiplied by the number of iterations, which equals the number of linear pieces that the one-dimensional line crosses within the search range of Algorithm 2. Although the number of polyhedral regions in the $n$-dimensional input space can be exponentially large, Algorithm 2 never enumerates them: by conditioning on the sufficient statistic of the nuisance parameter, the search is reduced to the one-dimensional line, and only the regions intersected by this line are visited. In the worst case, the number of crossed pieces can still grow exponentially with the number of units of the network. However, Hanin & Rolnick (2019) prove that for piecewise-linear networks whose weights and biases follow a distribution admitting a density, the expected number of linear regions crossed by a one-dimensional line grows only linearly with the total number of neurons; this is proven at initialization, and they empirically observe that the linear order is maintained during training. This result does not provide a formal complexity guarantee for our trained recursive diffusion architecture, but it offers theoretical support for the empirical observation that the number of crossed regions can grow much more slowly than the worst-case exponential bound. Our experiments in Figure 8 directly assess this behavior for the architectures considered in this study.

**Empirical scalability**  Specifically, we measured the number of intervals searched by Algorithm 2 on trained diffusion models under the same settings as in §5.1, while increasing the image size $n \in \{64, 256, 1024, 4096\}$ (the network size grows with $n$ accordingly). Figure 8a shows the results. While the image size increases 64-fold, the number of intervals grows only by a factor of less than four in both the independence and correlation settings, showing no sign of exponential growth.

**Runtime**   Figure 8b shows the computation time of the proposed method when changing the image size for the synthetic data, under the same settings as in §5.1. To optimize performance, we applied an acceleration technique that enables early termination once the $p$-value reaches sufficient precision; the details of this technique are described in Shiraishi et al. (2024a). Table 2 shows the computation times for the brain image dataset described in §5.2, where the times were averaged over 100 images each of brains with and without tumors, and the interval calculations for the $p$-value were performed in parallel using 48 cores. The computation time was 1100 seconds per image without tumors and 4220 seconds per image with tumors.

Table 2: Computation time for brain images using parallel processing across 48 cores.

| Image | Time (s) |
|---|---|
| Brain image without tumors | 1100 |
| Brain image with tumors | 4220 |

## E   Comparison methods for numerical experiments

We compared our proposed method with the following methods:

- `proposed`: The proposed method uses parametric programming.

- `w/o-pp`: The proposed method uses over-conditioning (without parametric programming). The $p$-value is calculated as

$$p_{\text{ablation}} = \mathbb{P}_{\text{H}_0}\left(|T(\boldsymbol{X}(Z), \boldsymbol{X}^{\text{ref}}(Z))| > |T(\boldsymbol{x}, \boldsymbol{x}^{\text{ref}})| \mid Z \in \mathcal{Z}^{\text{sub}}(\boldsymbol{a} + \boldsymbol{b}z^{\text{obs}})\right)$$

- `naive`: The naive method. This method uses a conventional $z$-test without any conditioning in (8).

- `bonferroni`: To control the type I error rate, this method applies the Bonferroni correction. Given that the total number of anomaly regions is $2^n$, the $p$-value is calculated as

$$p_{\text{bonferroni}} = \min(1, 2^n \cdot p_{\text{naive}}).$$

- `permutation`: This method uses a permutation test with the steps outlined below:

  - Calculate the observed test statistic $z^{\text{obs}}$ by applying the observed image $\boldsymbol{x}$ to the diffusion model.
  - For each $i = 1, \ldots, B$, compute the test statistic $z^{(i)}$ by applying the permuted image $\boldsymbol{X}^{(i)}$ to the diffusion model, where $B$ represents the total number of permutations, set to 1,000 in our experiments.

$$p_{\text{permutation}} = \frac{1}{B} \sum_{b \in [B]} \mathbf{1}\{|z^{(b)}| > |z^{\text{obs}}|\},$$

  where $\mathbf{1}\{\cdot\}$ denotes the indicator function.

## F   Architecture of the U-Net

Figure 9 shows the architecture of the U-Net used in our experiments. The U-Net has three skip connections, and the Encoder and Decoder blocks. For image sizes $n \in \{64, 256, 1024, 4096\}$, the corresponding spatial dimensions of images are $(1, d, d)$ where $d \in \{8, 16, 32, 64\}$.

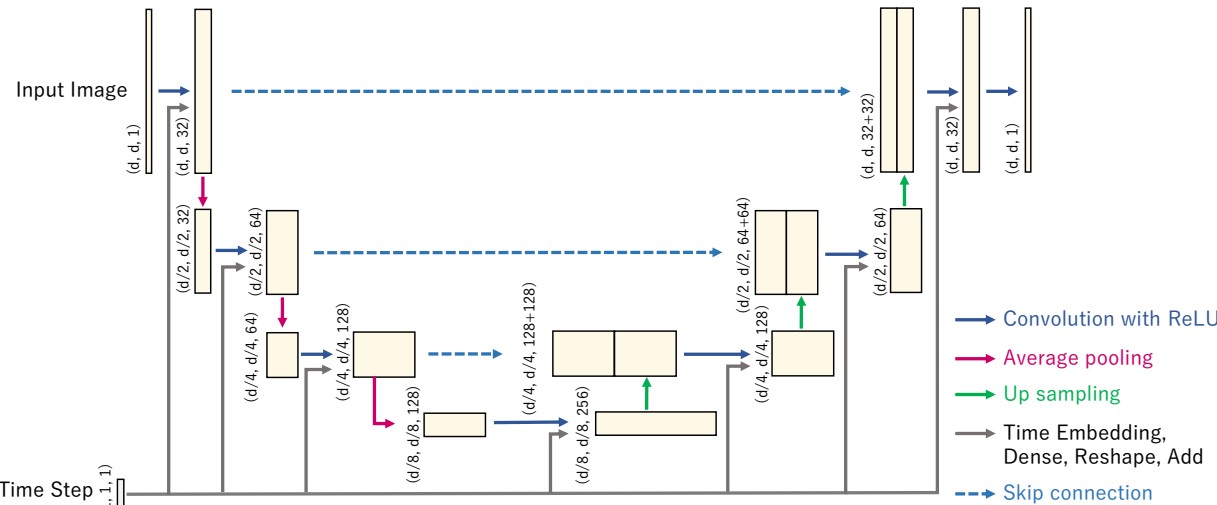

Figure 9: The architecture of the U-Net

## G Experimental settings

### G.1 Hyperparameter settings

Table 3 consolidates the hyperparameters used in our experiments: the numerical experiments in §5.1 and the real-world data experiments in §5.2 (see the following subsections for the dataset-specific settings). The threshold $\lambda$ and the kernel size of the averaging filter determine the anomalous region in (5), and $T$, $T'$, the number of sampling steps, and $\eta$ specify the reconstruction process of the diffusion model.

Table 3: Hyperparameters used in the experiments.

| Hyperparameter | Numerical (§5.1) | BraTS | MVTec AD |
|---|---|---|---|
| Threshold $\lambda$ | 0.8 | 0.6 | 1.0 (bottle, transistor), 1.2 (cable, grid) |
| Kernel size of the averaging filter | 3 | 3 | 3 |
| Total diffusion steps $T$ | 1000 | 1000 | 1000 |
| Initial step of the reverse process $T'$ | 460 | 300 | 300 |
| Number of sampling steps | 5 | 5 | 4 |
| Stochasticity $\eta$ | 1 | 1 | 1 |
| Significance level $\alpha$ | 0.05 | 0.05 | 0.05 |

### G.2 Experimental settings of Brain Tumor Segmentation 2023 dataset

We evaluate our method on T2-FLAIR MRI brain scans from the Brain Tumor Segmentation 2023 dataset (Karargyris et al., 2023; LaBella et al., 2023). T2-FLAIR MRI comprises 934 non-skull-stripped 3D scans with dimensions of $240 \times 240 \times 155$. From these scans, we extracted 2D $240 \times 240$ axial slices at axis 95, resized them to $64 \times 64$ pixels, and categorized them based on the anomaly annotations into 532 normal images (without tumors) and 402 abnormal images (with tumors). For each scan, we estimated the mean and variance from pixel values excluding both the non-brain regions and tumor regions identified in the ground truth, followed by standardization. We randomly selected 312 normal images for model training. The model was trained with $T = 1000$ and the initial time step of the reverse process was set at $T' = 300$, with reconstruction performed through 5 step samplings. We set the threshold $\lambda = 0.6$ and the kernel size of the averaging filter to 3. Note that, when testing images of the MRI brain scans, the non-brain regions are not treated as anomalous regions $\mathcal{M}_{\boldsymbol{X}}$.

### G.3 Experimental settings of MVTec AD dataset

We evaluate our method on the MVTec AD dataset (Bergmann et al., 2019), which consists of 15 object categories. Each category provides a training set of normal images and a test set containing both normal and abnormal images, with image resolutions ranging from $900 \times 900$ to $1024 \times 1024$ pixels. For our experiments, we select four categories (bottle, cable, grid, transistor) and resize all images to $128 \times 128$ pixels. For the type I error rate experiments, we randomly select 50 normal images from each category, and for the power experiments, we select 50 abnormal images per category. The diffusion model is trained on the remaining normal images with $T = 1000$ total diffusion steps, of which the first $T' = 300$ steps are used for reconstruction using 4 sampling steps. We apply an averaging filter with a kernel size of 3. We set the anomaly threshold $\lambda$ to 1.0 for bottle and transistor, and to 1.2 for cable and grid. Each image is standardized using the channel-wise mean and variance estimated from held-out normal images, which are excluded from the training, test, and reference sets. In the power experiments, we compute the intersection between the anomalous region detected by the diffusion model and the anomaly annotation. Since the images in the MVTec AD dataset are RGB, we redefine the image data as $\tilde{\boldsymbol{X}} \in \mathbb{R}^{hw \times 3}$, where $h$ and $w$ denote the image height and width. We then vectorize each image by

$$\boldsymbol{X} = \mathrm{vec}(\tilde{\boldsymbol{X}}) = (X_{1,1}, X_{1,2}, X_{1,3}, X_{2,1}, X_{2,2}, X_{2,3}, \ldots, X_{hw,1}, X_{hw,2}, X_{hw,3})^\top \in \mathbb{R}^n,$$

Similarly, we define the reference image $\tilde{\boldsymbol{X}}^{\mathrm{ref}} \in \mathbb{R}^{hw \times 3}$ as the average of the training images, and vectorize it in the same way as above.

$$\boldsymbol{X}^{\mathrm{ref}} = \mathrm{vec}(\tilde{\boldsymbol{X}}^{\mathrm{ref}}) = (\tilde{X}_{1,1}, \tilde{X}_{1,2}, \tilde{X}_{1,3}, \tilde{X}_{2,1}, \tilde{X}_{2,2}, \tilde{X}_{2,3}, \ldots, \tilde{X}^{\mathrm{ref}}_{hw,1}, \tilde{X}^{\mathrm{ref}}_{hw,2}, \tilde{X}^{\mathrm{ref}}_{hw,3})^\top \in \mathbb{R}^n$$

where $n = 3hw$, and accordingly redefine the test statistic in (7) as

$$T(\boldsymbol{X}, \boldsymbol{X}^{\mathrm{ref}}) = \frac{1}{|\mathcal{M}_{\boldsymbol{X}}|} \sum_{i \in \mathcal{M}_{\boldsymbol{x}}} \sum_{j \in [3]} \tilde{X}_{i,j} - \frac{1}{|\mathcal{M}_{\boldsymbol{X}}|} \sum_{i \in \mathcal{M}_{\boldsymbol{x}}} \sum_{j \in [3]} \tilde{X}^{\mathrm{ref}}_{i,j}$$

With this setup, our method can be applied in the same way as in §3.

## H   Accelerated reverse processes

Methods for accelerating the reverse process have been proposed in DDPM, DDIM (Song et al., 2022). When taking a strictly increasing subsequence $\tau$ from $\{1, \cdots, T\}$, it is possible to skip the sampling trajectory from $\mathbf{x}_{\tau_i}$ to $\mathbf{x}_{\tau_{i-1}}$. In this case, equations (2) and (4) can be rewritten as

$$\mathbf{x}_{\tau_{i-1}} = \sqrt{\alpha_{\tau_{i-1}}} \left( \frac{\mathbf{x}_{\tau_i} - \sqrt{1 - \alpha_{\tau_i}} \cdot \epsilon^{(\tau_i)}(\mathbf{x}_{\tau_i})}{\sqrt{\alpha_{\tau_i}}} \right) + \sqrt{1 - \alpha_{\tau_{i-1}} - \sigma^2_{\tau_i}} \cdot \epsilon^{(\tau_i)}_\theta(\mathbf{x}_{\tau_i}) + \sigma_{\tau_i} \epsilon_{\tau_i},$$

where

$$\sigma_{\tau_i} = \eta \sqrt{(1 - \alpha_{\tau_{i-1}})/(1 - \alpha_{\tau_i})} \sqrt{1 - \alpha_{\tau_i}/\alpha_{\tau_{i-1}}}.$$

Therefore, piecewise-linearity is preserved, making the proposed method DAL-Test applicable.

## I   Robustness of the proposed method

To evaluate the robustness of the proposed method against deviations from the Gaussian assumption, we applied our method without any modification (i.e., assuming Gaussianity) to data generated from various non-Gaussian distribution families with different levels of deviation from the standard normal distribution $\mathcal{N}(0, 1)$. We considered the following non-Gaussian distributions with a 1-Wasserstein distance $d \in \{0.01, 0.02, 0.03, 0.04\}$ from $\mathcal{N}(0, 1)$:

- Skew normal distribution family (SND).

- Exponentially modified gaussian distribution family (EMG).

- Generalized normal distribution family (GND) with a shape parameter $\beta$. This distribution family can be steeper than the normal distribution (i.e., $\beta < 2$).

- Student's $t$-distribution family ($t$-distribution).

Note that these distributions are standardized in the experiments. Figure 10 shows the probability density functions for distributions from each family, such that the $d$ is set to 0.04. The significance levels $\alpha$ were set to 0.05 and 0.10, and the image size was set to 256. Figure 11 shows the results of the robustness experiments.

The `proposed` method maintains good performance on the type I error rate for all the considered distribution families and all distances $d$, indicating that the proposed method is robust against moderate deviations from the Gaussian assumption.

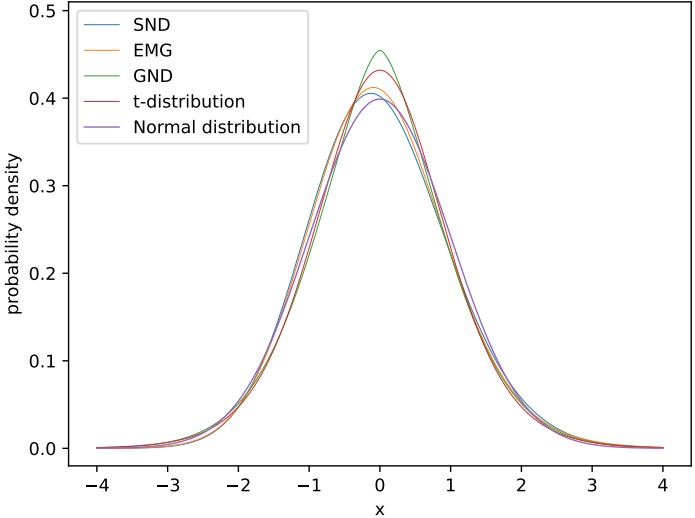

Figure 10: Non-Gaussian distributions with $d = 0.04$

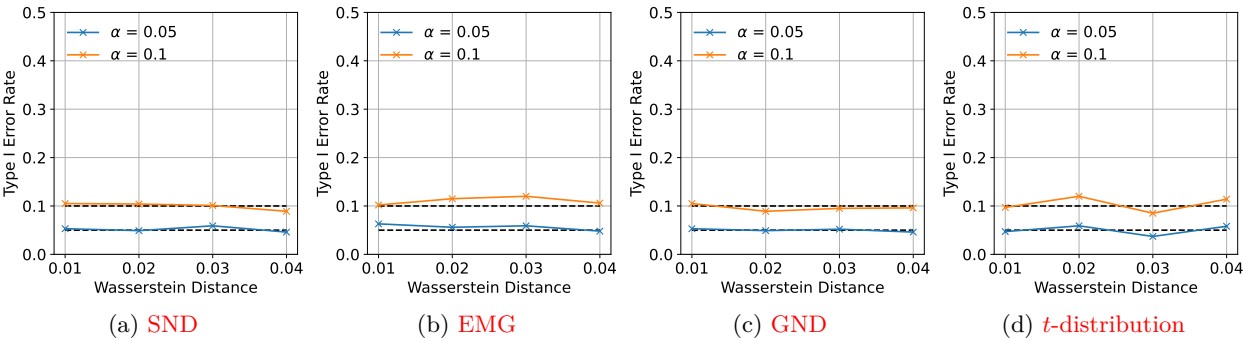

(a) SND          (b) EMG          (c) GND          (d) $t$-distribution

Figure 11: Type I Error Rate for Non-Gaussian distribution families

