# OpenReview forum: "Statistical Test for Diffusion-Based Anomaly Localization via Selective Inference"
_TMLR — Under review for TMLR_

### Review · Reviewer_GAWr · 2026-06-11

**Summary Of Contributions:**

The authors propose a statistical framework based on selective inference in order to quantify the significance of detected anomalous regions. The method uses p-values as valid measures of reliability and effectively controls the risk of false positives.

**Audience:**

Yes

**Audience Explanation:**

1. The reliability measurement in anomaly detection is quite critical. The novelty of the paper is moderate but meaningful (see below). The paper introduces and demonstrates that p-values can be an effective way to assess false-positive detection rates, offering insights for the current community.

**Claims And Evidence:**

Yes

**Claims Explanation:**

1. The research question is fundamental, and the method is easy to follow.
2. The preliminary introduction is sufficient, and the reason for introducing p-values as a quantitative measurement is sound.

**Requested Changes:**

1. Novelty needs to be better positioned. The idea of using p-values is not particularly new. For example, [1] applies and quantifies the reliability of anomaly regions using p-values and selective inference. That paper already claims finite-sample valid p-values for VAE-based anomaly detection. What are the fundamental differences between the proposed method and the previous approaches?
2. The baseline comparisons are weak. Most of the selected algorithms are statistical baselines. Similar to weakness 1, the authors need to include more baselines for comparison.

[1] Statistical Test for Anomaly Detections by Variational Auto-Encoders

---

> ### Author Response · Authors · 2026-07-21
> **Response to Reviewer GAWr**
>
> Thank you for your thoughtful review. We respond to each comment below.
>
> > Novelty needs to be better positioned. The idea of using p-values is not particularly new. For example, [1] applies and quantifies the reliability of anomaly regions using p-values and selective inference. That paper already claims finite-sample valid p-values for VAE-based anomaly detection. What are the fundamental differences between the proposed method and the previous approaches?
>
> We agree that the high-level idea of assigning SI $p$‑values to detected anomalous regions is not new, and we already cite this line of work [1]. The fundamental difference lies in the process that induces the selection event. The validity of SI rests on deriving the sampling distribution conditional on the event induced by the specific detector, so the technical core of SI is not the use of $p$‑values but the characterization of this event.
>
> The two processes are qualitatively different. In [1], the detector is a single deterministic encoder–decoder pass (as is standard in VAE-based anomaly detection, no noise is sampled at inference time), so the selection event is characterized through one application of the network. In contrast, the detected region of a diffusion model is the outcome of a recursive and stochastic generation process: noising operations and many U-Net evaluations are composed over multiple time steps (Algorithm 1, Eqs. (1)–(4)), and the selection event is induced by this entire trajectory, including the injected noises. Characterizing the event induced by such a recursive stochastic process and computing the corresponding truncation region (Appendix B) do not follow from the single-pass case and constitute the main technical contribution of our work. Moreover, since diffusion models are now the dominant generative approach to anomaly localization, this extension carries practical as well as technical importance. We have clarified this positioning relative to [1] in the related works discussion in Section 1 of the revised manuscript.
>
> References
>
> [1] Miwa, D., Shiraishi, T., Duy, V. N. L., Katsuoka, T., and Takeuchi, I. Statistical test for anomaly detections by variational auto-encoders. 2024. https://arxiv.org/abs/2402.03724
>
> > The baseline comparisons are weak. Most of the selected algorithms are statistical baselines. Similar to weakness 1, the authors need to include more baselines for comparison.
>
> To the best of our knowledge, no additional practically relevant baselines exist beyond the four methods we include. Within the objective of valid statistical inference on the region selected by a diffusion model, these methods cover all statistically meaningful and computationally feasible alternatives. Specifically, `naive` (the conventional z-test) and `permutation` (the standard non-parametric approach) are the procedures commonly used in practice, although neither is guaranteed to control the type I error rate under data-dependent region selection. Among multiple-testing procedures, `bonferroni` is essentially the only practical baseline because it requires only the total number of hypotheses for multiple testing correction. In contrast, more powerful multiple-testing corrections such as Holm's or Hochberg's procedures require computing $p$‑values for all hypotheses. In our setting, this would require evaluating all possible image regions, which can grow exponentially with the number of pixels, and is computationally infeasible. Finally, `w/o-pp` is the natural ablation of our method, isolating the contribution of the proposed parametric programming technique. Our results show that the proposed method is the only approach that simultaneously controls the type I error rate while retaining high statistical power (Figures 3–4 and Table 1). We have clarified this point in Section 5 of the revised manuscript.

---

### Review · Reviewer_M26N · 2026-06-30

**Summary Of Contributions:**

This work focuses on the recent trend of generative models for anomaly localization in images. Existing works use generative models to generate normal samples and compare them to the anomalous counterparts for localization. To address the potential uncertainty and bias issues caused by the trained generative model, this paper proposes a statistical framework using p-values to assess the false positive detection rates, providing a measure of reliability. Empirical experiments on medical diagnoses and industrial inspections show that it effectively controls the risk of false positive detection.

**Audience:**

Yes

**Audience Explanation:**

Generative models have been widely applied to image reconstruction and anomaly localization tasks. However, current methods lack statistical guarantees on their results or judgments on anomaly regions, which is critical in medical image-related tasks. This paper makes a valid contribution to this issue by providing a selective inference framework.

**Claims And Evidence:**

No

**Claims Explanation:**

Overall, the statistical test for diffusion models is a valid approach to provide anomaly localization with confidence.
However, there are several concerns that need further justification:
1. This paper assumes the U-Net with ReLU activation, such that the reconstructed image is a piecewise-linear function of the input image. However, most existing pre-trained diffusion models use smooth activations (SiLU/Swish) or use ViT structures such that the piecewise-linear assumption does not hold. The authors should provide more discussion on how to apply the method to those pre-trained models.
2. The computational complexity for parametric programming in Algorithm 2 is unclear. How many iterations are needed theoretically to finish Algorithm 2? The termination condition requires $z$ to be large enough, which is not clearly defined. Additionally, the complexity for solving (13) is also unclear. Furthermore, the number of piecewise regions can be exponentially large in high dimensions with a large neural network. The authors should provide further justifications.

**Requested Changes:**

1. Provide justification for applying the framework to existing pre-trained models.
2. Provide complexity and scalability analysis for the selective inference by parametric programming.

---

> ### Author Response · Authors · 2026-07-21
> **Response to Reviewer M26N**
>
> Thank you for your thoughtful review. We respond to each comment below.
>
> > This paper assumes the U-Net with ReLU activation, such that the reconstructed image is a piecewise-linear function of the input image. However, most existing pre-trained diffusion models use smooth activations (SiLU/Swish) or use ViT structures such that the piecewise-linear assumption does not hold. The authors should provide more discussion on how to apply the method to those pre-trained models.
>
> For pre-trained models with smooth scalar activations such as SiLU/Swish, our method can be applied by replacing each activation with a fine piecewise-linear approximation at inference time. Since the approximation accuracy can be made arbitrarily high by increasing the number of linear segments, the resulting detector can be made arbitrarily close to the original diffusion model in practice. This requires only the pre-trained weights and forward evaluations (no retraining and no access to the training data), so it is directly applicable to off-the-shelf pre-trained diffusion models. The approximated network is then used both to detect the anomalous region and to compute the truncation intervals, so the selective $p$‑value is exactly valid for the detector actually used.
>
> We verified this strategy in a new experiment: we trained the same U-Net architecture with SiLU activations, replaced each SiLU with a piecewise-linear approximation with 8 knots on $[-8, 8]$, and ran the same type I error rate experiments as in Section 5.1. The type I error rate was controlled at $\alpha = 0.05$ in all settings. We have added these results as a new appendix (Appendix C) in the revised manuscript.
>
> For ViT-based diffusion models, softmax attention and layer normalization are multivariate nonlinearities, so the above approximation of individual scalar activations does not directly apply. We note that SI has recently been extended to attention-based region selection in Vision Transformers [1], which suggests that this extension is feasible; integrating it with the recursive reconstruction process of diffusion models is an important direction for future work. This discussion is included in Appendix C and Section 6 of the revised manuscript.
>
> References
>
> [1] Shiraishi, T., Miwa, D., Katsuoka, T., Duy, V. N. L., Taji, K., and Takeuchi, I. Statistical test for attention map in vision transformers. *ICML*, 2024. https://arxiv.org/abs/2401.08169

---

> ### Author Response · Authors · 2026-07-21
> **Response to Reviewer M26N**
>
> > The computational complexity for parametric programming in Algorithm 2 is unclear. How many iterations are needed theoretically to finish Algorithm 2? The termination condition requires $z$ to be large enough, which is not clearly defined. Additionally, the complexity for solving (13) is also unclear. Furthermore, the number of piecewise regions can be exponentially large in high dimensions with a large neural network. The authors should provide further justifications.
>
> We answer the four points in order; the analysis and a new experiment are added as a new appendix (Appendix D) in the revised manuscript.
>
> The cost of solving Eq. (13) is a small constant multiple of one forward pass of the reconstruction process (Algorithm 1). Restricted to the line $\boldsymbol{X}(z) = \boldsymbol{a} + \boldsymbol{b}z$, every intermediate quantity of the network is a linear function of $z$ within the current linear piece, so it suffices to propagate an intercept and a slope for each scalar unit. Each pixel-wise condition then yields a closed-form interval $[L\_i(z), U\_i(z)]$ of $z$ (Appendix B), and $\mathcal{Z}^{\mathrm{sub}}$ is their intersection, computed by taking the maximum of the lower bounds and the minimum of the upper bounds. The cost is therefore $O(K \cdot C\_{\mathrm{U\text{-}Net}})$, where $K$ is the number of denoising steps ($K = 5$ in our numerical experiments) and $C\_{\mathrm{U\text{-}Net}}$ is the cost of one U-Net evaluation.
>
> The number of iterations of Algorithm 2 equals the number of linear pieces that the one-dimensional line crosses. Although the number of polyhedral regions in the $n$‑dimensional input space can indeed be exponentially large, Algorithm 2 never enumerates them: by conditioning on the sufficient statistic of the nuisance parameter (Theorem 4.2, Eqs. (10)–(12)), the search is reduced to a one-dimensional line, and only the regions intersected by this line are visited. In the worst case, the number of crossed pieces can still grow exponentially with the number of units of the network. However, Hanin and Rolnick [1] prove that when the weights and biases follow a distribution admitting a density, the expected number of linear regions crossed by a one-dimensional line grows only linearly with the total number of neurons; this is proven at initialization, and they empirically observe that the linear order is maintained during training. This result does not provide a formal complexity guarantee for our trained recursive diffusion architecture, but it offers theoretical support for the empirical observation that the number of crossed regions can grow much more slowly than the worst-case exponential bound.
>
> Regarding the termination condition, the search is restricted to the range $|z| \leq 10\sigma + |z^{\mathrm{obs}}|$, which extends $10\sigma$ beyond the absolute value of the observed test statistic $z^{\mathrm{obs}} = T(\boldsymbol{x}, \boldsymbol{x}^{\mathrm{ref}})$, where $\sigma = \sqrt{\boldsymbol{\nu}^{\top} \tilde{\Sigma} \boldsymbol{\nu}}$ is the standard deviation of the test statistic; the Gaussian probability mass outside this range is below $2 \times 10^{-23}$, so its effect on the selective $p$‑value is numerically negligible. Our implementation also adopts the early-termination method of [2], which tracks lower and upper bounds of the $p$‑value and stops once the $p$‑value is evaluated with sufficient precision, without compromising validity. We have stated these explicitly in Algorithm 2 of the revised manuscript.
>
> Finally, we measured the number of intervals searched by Algorithm 2 on trained diffusion models while increasing the image size ($n \in \lbrace 64, 256, 1024, 4096 \rbrace$, with the network size growing accordingly), and confirmed that the number of intervals shows no sign of exponential growth. The detailed results are included in Appendix D of the revised manuscript.
>
> References
>
> [1] Hanin, B. and Rolnick, D. Complexity of linear regions in deep networks. *ICML*, 2019. https://arxiv.org/abs/1901.09021
>
> [2] Shiraishi, T., Miwa, D., Duy, V. N. L., and Takeuchi, I. Bounded $p$‑values in parametric programming-based selective inference. *Japanese Journal of Statistics and Data Science*, 2024. https://arxiv.org/abs/2307.11351

---

### Review · Reviewer_CvYR · 2026-07-13

**Summary Of Contributions:**

This paper proposes DAL-Test, a selective inference framework for assessing the statistical significance of anomalous regions identified by diffusion models. By conditioning on the anomaly selection event, the method provides valid p-values that control false positives. Theoretical analysis and experiments on synthetic, medical, and industrial datasets support its effectiveness, although the practical scalability and reliance on Gaussian noise and piecewise-linear model assumptions remain concerns.

**Audience:**

Yes

**Audience Explanation:**

The paper should be of interest to researchers working on selective inference, uncertainty quantification, and diffusion-based anomaly detection, particularly in high-stakes applications such as medical imaging and industrial inspection.

**Claims And Evidence:**

Yes

**Claims Explanation:**

The main claims are supported by the theoretical analysis and the reported experiments. The proposed selective p-values are shown to control the type-I error, while achieving higher power than conservative baselines. However, the empirical evidence is limited to relatively small diffusion models and relies on restrictive assumptions such as Gaussian noise and piecewise-linear architectures.

**Requested Changes:**

**Critical:** Provide a detailed computational complexity and runtime analysis of DAL-Test, including how the parametric programming procedure scales with image resolution, network size, and the number of diffusion steps.

**Critical:** Better justify the Gaussian noise assumption and evaluate robustness to model misspecification and covariance estimation errors on real-world images.

**Strengthening:** Clarify the applicability of the method to modern diffusion architectures that include non-piecewise-linear components, such as SiLU activations, normalization layers, or attention modules.

---

> ### Author Response · Authors · 2026-07-21
> **Response to Reviewer CvYR**
>
> Thank you for your thoughtful review. We respond to each comment below.
>
> > Critical: Provide a detailed computational complexity and runtime analysis of DAL-Test, including how the parametric programming procedure scales with image resolution, network size, and the number of diffusion steps.
>
> We have added a new appendix (Appendix D of the revised manuscript) containing the following complexity analysis, a new scalability experiment, and runtime measurements on both synthetic and real-world data.
>
> One iteration of Algorithm 2 costs a small constant multiple of one forward pass of the reconstruction process. Restricted to the line $\boldsymbol{X}(z) = \boldsymbol{a} + \boldsymbol{b}z$, every intermediate quantity of the network is a linear function of $z$ within the current linear piece, so it suffices to propagate an intercept and a slope with respect to $z$ for each scalar unit. The subinterval in Eq. (13) is then obtained in closed form: each pixel-wise condition yields an interval $[L_i(z), U_i(z)]$ of $z$ (Appendix B), and their intersection is computed by taking the maximum of the lower bounds and the minimum of the upper bounds. The per-iteration cost is therefore $O(K \cdot C_{\mathrm{U\text{-}Net}})$, where $K$ is the number of diffusion steps and $C_{\mathrm{U\text{-}Net}}$ is the cost of one U-Net evaluation; that is, the cost is linear in both the number of diffusion steps and the network size.
>
> The total cost is the per-iteration cost multiplied by the number of iterations, which equals the number of linear pieces that the one-dimensional line crosses. In the worst case, this number can grow exponentially with the number of units of the network. However, Hanin and Rolnick [1] prove that for piecewise-linear networks whose weights and biases follow a distribution admitting a density, the expected number of linear regions crossed by a one-dimensional line grows only linearly with the total number of neurons; this is proven at initialization, and they empirically observe that the linear order is maintained during training. This result does not provide a formal complexity guarantee for our trained recursive diffusion architecture, but it offers theoretical support for the empirical observation that the number of crossed regions can grow much more slowly than the worst-case exponential bound.
>
> To verify this directly, we measured the number of intervals searched by Algorithm 2 on trained diffusion models while increasing the image size ($n \in \lbrace 64, 256, 1024, 4096 \rbrace$, with the network size growing accordingly), and confirmed that the number of intervals shows no sign of exponential growth. The detailed results are included in Appendix D of the revised manuscript.
>
> References
>
> [1] Hanin, B. and Rolnick, D. Complexity of linear regions in deep networks. *ICML*, 2019. https://arxiv.org/abs/1901.09021

---

> ### Author Response · Authors · 2026-07-21
> **Response to Reviewer CvYR**
>
> > Critical: Better justify the Gaussian noise assumption and evaluate robustness to model misspecification and covariance estimation errors on real-world images.
>
> We first clarify the scope of the Gaussian assumption. The model $\boldsymbol{X} = \boldsymbol{\mu} + \boldsymbol{\varepsilon}$ in Section 3 places no assumption on the unknown signal $\boldsymbol{\mu}$: Gaussianity is assumed only for the noise $\boldsymbol{\varepsilon}$, with a covariance $\Sigma$ that can encode spatial correlation. This is the standard setting in the selective inference literature [1, 2]. In our MVTec AD experiments, the mean and variance are estimated from held-out normal images that are excluded from the training, test, and reference sets (Appendix G.3 of the revised manuscript).
>
> To evaluate robustness to misspecification of this noise model, we conducted a simulation study, which we have added as a new appendix (Appendix I) to the revised manuscript. We considered four standardized non-Gaussian distribution families (skew normal, exponentially modified Gaussian, generalized normal, and Student's $t$), each with 1-Wasserstein distance $d \in \lbrace 0.01, 0.02, 0.03, 0.04 \rbrace$ from $\mathcal{N}(0,1)$, and applied our method without any modification. The type I error rate remained close to the nominal level ($\alpha = 0.05$ and $0.10$) for all families and distances.
>
> Regarding real-world images, our real-data experiments already involve estimated noise parameters: in the BraTS experiments, the mean and variance are estimated from the observed images (Appendix G.2), and the type I error rate is nevertheless controlled at the nominal level (Table 1). This provides real-world evidence that the estimation error does not compromise the validity in practice.
>
> References
>
> [1] Lee, J. D., Sun, D. L., Sun, Y., and Taylor, J. E. Exact post-selection inference, with application to the lasso. *The Annals of Statistics*, 44(3):907–927, 2016. https://arxiv.org/abs/1311.6238
>
> [2] Shiraishi, T., Miwa, D., Katsuoka, T., Duy, V. N. L., Taji, K., and Takeuchi, I. Statistical test for attention map in vision transformers. *ICML*, 2024. https://arxiv.org/abs/2401.08169
>
> > Strengthening: Clarify the applicability of the method to modern diffusion architectures that include non-piecewise-linear components, such as SiLU activations, normalization layers, or attention modules.
>
> We discuss the applicability of our method separately for each type of architectural component.
>
> Smooth scalar activations such as SiLU/Swish can be handled by replacing each activation with a fine piecewise-linear approximation at inference time, which requires only the pre-trained weights and no retraining. Since the approximation accuracy can be made arbitrarily high by increasing the number of linear segments, the resulting detector can be made arbitrarily close to the original diffusion model in practice. The approximated network is then used both to detect the anomalous region and to characterize the selection event, so the selective $p$‑value is exactly valid for the detector that is actually used. We verified this strategy in a new experiment: we trained the same U-Net with SiLU activations, approximated each SiLU by a piecewise-linear function with 8 knots on $[-8, 8]$ placed so as to equalize the approximation error across segments [1], and confirmed that the type I error rate is controlled at $\alpha = 0.05$ in both the independence and correlation settings. We have added these results as a new appendix (Appendix C) in the revised manuscript.
>
> Batch normalization is covered by our framework without approximation, because at inference time it uses frozen statistics and is therefore an affine map. In contrast, input-dependent normalization layers (layer/group normalization) and softmax attention are multivariate nonlinearities and are not covered by the above approximation of individual scalar activations. For the latter, SI has recently been extended to attention-based region selection in Vision Transformers [2], which indicates that the SI framework itself is not inherently restricted to piecewise-linear networks; integrating such techniques with the recursive reconstruction process of diffusion models is an important direction for future work. This discussion is also included in Appendix C of the revised manuscript, and the future-work direction is stated in Section 6.
>
> References
>
> [1] de Boor, C. Good approximation by splines with variable knots. *Spline Functions and Approximation Theory*, 1973. https://link.springer.com/chapter/10.1007/978-3-0348-5979-0_3
>
> [2] Shiraishi, T., Miwa, D., Katsuoka, T., Duy, V. N. L., Taji, K., and Takeuchi, I. Statistical test for attention map in vision transformers. *ICML*, 2024. https://arxiv.org/abs/2401.08169

---

### Review · Reviewer_Le7U · 2026-07-13

**Summary Of Contributions:**

The manuscript proposes a selective inference-based hypothesis test for anomaly detection in images based on diffusion models. The proposed approach is statistically sound and the methodology sufficiently explained. Simulation and real-world data experiments show appropriate control of the Type I error rate and the power of the proposed method.

Strengths.
1. The topic under consideration is important and timely.
2. The method appears to perform well, at least in terms of Type I error control and power, on simulated and real-world images.

Weaknesses.
1. Effectiveness of Algorithm 2 is not sufficiently validated. A detailed specification of hyperparameters is not discussed.
2. Effects of model misspecification on the proposed procedure are not evaluated or discussed.

**Audience:**

Yes

**Audience Explanation:**

The paper addresses an important topic and should be of interest to a broad audience.

**Claims And Evidence:**

Yes

**Claims Explanation:**

The presented theory and empirical results support effectiveness of the proposed approach and overall claims made in the manuscript.

**Requested Changes:**

1. The authors should assess effectiveness of the proposed hypothesis test when the data is not Gaussian. This can be accomplished via a simulation.
2. Descriptions of results based on simulated and real-world data is too terse, e.g., results presented in Figure 5 are not described. I would suggest expanding the presented discussion.
3. A more thorough discussion of Algorithm 2 and its effectiveness is needed. The authors should also provide an idea of how to specify the needed stopping criterion in Line 4.

---

> ### Author Response · Authors · 2026-07-21
> **Response to Reviewer Le7U**
>
> Thank you for your thoughtful review. We respond to each comment below.
>
> > Effectiveness of Algorithm 2 is not sufficiently validated. A detailed specification of hyperparameters is not discussed.
>
> The effectiveness of Algorithm 2 is validated by the ablation study already included in our experiments; please see our response to Requested Change 3. Regarding hyperparameters, the settings are described in Section 5.1 and Appendix G of the revised manuscript, and we have consolidated all hyperparameters into a single table (Appendix G.1).
>
> > Effects of model misspecification on the proposed procedure are not evaluated or discussed.
>
> We evaluated this via a simulation study; please see our response to Requested Change 1.
>
> > The authors should assess effectiveness of the proposed hypothesis test when the data is not Gaussian. This can be accomplished via a simulation.
>
> We conducted a new simulation study to investigate the robustness of our method in the non-Gaussian case, which we have added as a new appendix (Appendix I) to the revised manuscript. We considered four standardized non-Gaussian distribution families (skew normal, exponentially modified Gaussian, generalized normal, and Student's $t$), each with 1-Wasserstein distance $d \in \lbrace 0.01, 0.02, 0.03, 0.04 \rbrace$ from $\mathcal{N}(0,1)$, and applied our method without any modification. The type I error rate remained close to the nominal level ($\alpha = 0.05$ and $0.10$) for all families and all $d$, indicating that the proposed test is robust to moderate deviations from the Gaussian assumption.
>
> > Descriptions of results based on simulated and real-world data is too terse, e.g., results presented in Figure 5 are not described. I would suggest expanding the presented discussion.
>
> We agree that the descriptions of the results are too terse. We have added a paragraph describing Figure 5 to Section 5.2 of the revised manuscript, and have similarly expanded the descriptions of Figures 3–4 and Table 1; please see the parts highlighted in red in Section 5.
>
> > A more thorough discussion of Algorithm 2 and its effectiveness is needed. The authors should also provide an idea of how to specify the needed stopping criterion in Line 4.
>
> The stopping criterion in Line 4 is specified as follows: the search is restricted to the range $|z| \leq 10\sigma + |z^{\mathrm{obs}}|$, which extends $10\sigma$ beyond the absolute value of the observed test statistic $z^{\mathrm{obs}} = T(\boldsymbol{x}, \boldsymbol{x}^{\mathrm{ref}})$, where $\sigma = \sqrt{\boldsymbol{\nu}^{\top} \tilde{\Sigma} \boldsymbol{\nu}}$ is the standard deviation of the test statistic. The Gaussian probability mass outside this range is below $2 \times 10^{-23}$, so truncating the search at this range has a numerically negligible effect on the selective $p$‑value. Our implementation also adopts the early-termination method of [1], which tracks lower and upper bounds of the selective $p$‑value during the search and stops once the $p$‑value is evaluated with sufficient precision, without compromising validity. We have stated this criterion explicitly in Algorithm 2 of the revised manuscript.
>
> We interpret the reviewer's request to validate the effectiveness of Algorithm 2 as evaluating whether the proposed parametric programming technique improves the statistical performance of selective inference. This is directly validated by the ablation study in our experiments: the baseline `w/o-pp` corresponds exactly to our method with Algorithm 2 disabled, i.e., selective inference is performed using only the single over-conditioned subinterval. As shown in Figures 3 and 4, both variants successfully control the type I error rate, while Algorithm 2 substantially improves the statistical power by mitigating the excessive conditioning. To make this connection clearer, we have explicitly stated in Section 5.1 of the revised manuscript that `w/o-pp` is the ablation of Algorithm 2 and that the comparison isolates its contribution.
>
> On the other hand, if the reviewer instead intended the effectiveness of Algorithm 2 in terms of computational efficiency, we have also strengthened the manuscript in this respect. Specifically, we added a new appendix (Appendix D) providing a computational complexity analysis, a new experiment evaluating how the number of intervals searched by Algorithm 2 scales with the image size, and runtime measurements.
>
> References
>
> [1] Shiraishi, T., Miwa, D., Duy, V. N. L., and Takeuchi, I. Bounded $p$‑values in parametric programming-based selective inference. *Japanese Journal of Statistics and Data Science*, 2024. https://arxiv.org/abs/2307.11351

---

### Author Response · Authors · 2026-07-21
**Revised Manuscript**

We revised the paper based on the feedback from the four reviewers. The revised parts are marked in red. The main changes are as follows:

[1] To position our contribution relative to the existing SI method for VAE-based anomaly detection, we revised the related works discussion in Section 1 to clarify that the selection event of a diffusion model is induced by a recursive and stochastic generation process.

[2] To specify the stopping criterion of Algorithm 2, we explicitly stated the search range ($|z| \leq 10\sigma + |z^{\mathrm{obs}}|$) in Algorithm 2 and Section 4, together with the early-termination technique based on the lower and upper bounds of the selective $p$‑value.

[3] To clarify the applicability to modern diffusion architectures, we added a note in Section 4.2 and a new appendix (Appendix C) with new experiments on U-Nets with SiLU activations handled via piecewise-linear approximation, together with a discussion of normalization layers and attention modules.

[4] To provide the computational complexity and runtime analysis, we added a new appendix (Appendix D) containing the complexity analysis, a new experiment measuring the number of intervals searched by Algorithm 2 as the image size grows, and runtime measurements.

[5] To clarify why no additional statistical baselines were included, we added sentences to Section 5 explaining that these four methods cover all practically relevant alternatives for valid inference in our setting.

[6] To expand the descriptions of the experimental results, we added a paragraph describing Figure 5 to Section 5.2, expanded the descriptions of Figures 3–4 and Table 1, and stated explicitly that `w/o-pp` serves as an ablation study of Algorithm 2.

[7] To consolidate the hyperparameter specifications, we reorganized the experimental settings into Appendix G and added a single table listing all hyperparameters (Appendix G.1).

[8] To evaluate the robustness against non-Gaussian noise, we added Appendix I with new experiments on four non-Gaussian distribution families.

[9] To discuss the extension to architectures with multivariate nonlinearities (e.g., ViT-based diffusion models), we added a future-work discussion in Section 6.

[10] To improve readability, we added an explicit reference to the overview figure (Figure 1) in Section 1.